# ICESat laser altimetry over small mountain glaciers

D. Treichler and A. Kääb

Institute of Geosciences, University of Oslo, P.O. box 1047, 0316 Oslo, Norway

*Correspondence to*: D. Treichler (desiree.treichler@geo.uio.no)

**Abstract.** Using sparsely glaciated southern Norway as a case study, we assess the potential and limitations of ICESat laser altimetry for analysing regional glacier elevation change in rough mountain terrain. Differences between ICESat GLAS elevations and reference elevation data are plotted over time to derive a glacier surface elevation trend for the ICESat acquisition period 2003–2008. We find spatially varying biases between ICESat and three tested digital elevation models (DEMs): the Norwegian national DEM, SRTM DEM and a high resolution LiDAR DEM. For regional glacier elevation change, the spatial inconsistency of reference DEMs – a result of spatio-temporal merging – has the potential to significantly affect or dilute trends. Elevation uncertainties of all three tested DEMs exceed ICESat elevation uncertainty by an order of magnitude, and are thus limiting the accuracy of the method, rather than ICESat uncertainty. ICESat matches glacier size distribution of the study area well and measures also small ice patches not commonly monitored in-situ. The sample is large enough for spatial and thematic subsetting. Vertical offsets to ICESat elevations vary for different glaciers in southern Norway due to spatially inconsistent reference DEM age. We introduce a per-glacier correction that removes these spatially varying offsets, and considerably increases trend significance. Only after application of this correction do also individual campaigns fit to observed in-situ glacier mass balance. Our correction has the potential to improve glacier trend significance also for other causes of spatially varying vertical offsets, for instance due to radar penetration into ice and snow for the SRTM DEM, or as a consequence from mosaicking and merging that is common for national or global DEMs After correction of reference elevation bias, we find that ICESat provides a robust and realistic estimate of a moderately negative glacier mass balance of around -0.36m +/-0.07 ice per year. This regional estimate agrees well with the heterogeneous but overall negative in-situ glacier mass balance observed in the area.

## 1 Introduction

The role of mountain glaciers and snow as source for drinking water, irrigation and hydropower is getting increasing attention, not least due to the significant population increase and economic development in a number of mountain regions and surrounding lowlands (Jansson et al., 2003; Viviroli et al., 2007). Retreat of mountain glaciers is also a major cause of eustatic sea level rise (Gardner et al., 2013). But the response of some large glacierised systems to climatic changes is still poorly quantified, especially in regions with large climatic variability. The glacier regions least represented in long-term in-situ glacier monitoring programmes are those with largest ice volumes (Zemp et al., 2015), which are less inhabited, difficult

to access, and therefore not well studied. Regional estimates of ice loss recently gained importance, not least for assessing the current and future contribution of water stored in land ice masses to sea level rise (Gardner et al., 2013; Jacob et al., 2012; Marzeion et al., 2012; Radić et al., 2014; Radić and Hock, 2011) and for quantifying current runoff contribution from glacier imbalance (Kääb et al., 2015) or changes in the upstream cryosphere (e.g. Bliss et al., 2014; Immerzeel et al., 2010).

Remotely sensed data is of special value in remote mountain regions where measurements such as in-situ mass balance measurements are sparse or lacking completely.

Elevation data from Geoscience Laser Altimeter System (GLAS) on board the NASA Ice, Cloud and land Elevation Satellite (ICESat) provides likely the most consistent global elevation measurement currently available (Nuth and Kääb, 2011). The use of this data to derive thickness changes of Arctic ice caps is well established (Nuth et al., 2010; Moholdt et al., 2010;

Bolch et al., 2013; Nilsson et al., 2015; Slobbe et al., 2008). Kääb et al. (2012) have shown that, when combined with reference heights from a digital elevation model (DEM), ICESat data can successfully be used to derive regional-scale glacier mass balance even in rough topographies as the Himalayas. Subsequently, ICESat's elevation measures combined with the Shuttle Radar Topography Mission (SRTM) DEM were used to estimate sea level rise contributions from mountain glaciers globally (Gardner et al., 2013), regionally in High Mountain Asia (Neckel et al., 2014; Kääb et al., 2015), and even

for local glacier mass balance studies in the Kunlun Shan (Ke et al., 2015) and the Alps (Kropáček et al., 2014).

The increased public interest in glacier retreat, not least due to its effects on water resources stored in mountain glaciers, requires that the performance of ICESat over such terrain is carefully evaluated and associated error sources well characterised. This is especially important given that using ICESat data over mountain topography is at (or even exceeds) the limits of what the mission was designed for. As a case study for this purpose we chose the mountains of southern Norway.

With its comparably small and sparse glaciers, situated within a varied topographic setting of both steep and gentle mountains, we consider the region as a representative case for the limits of applicability of ICESat data for analysing changes of mountain glaciers. In contrast to large, remote areas like High Mountain Asia, the climatic framework and glacier responses are relatively well known and measured in southern Norway, and accurate, up-to-date glacier masks and a high-resolution reference DEM are available.

Specifically, we aim to address the following questions in our study:

- What prerequisites and conditions need to be fulfilled to make ICESat-derived elevation changes over a certain area a valid method to assess glacier volume changes?
- Is the ICESat track density high enough for the sparse glacier cover in the study region, and are the point samples along

ICESat profiles representative of the whole glacier population in southern Norway?

- Can a realistic elevation trend be retrieved for the years 2003–2009 (glacier volume loss), and is it possible to detect climate-related patterns, namely the spatial transition from maritime towards more continental glaciers with increasing distance to the coast?

- What is the minimum region size with respect to glacier density for ICESat GLAS data to ensure statistically significant results? Are realistic annual glacier thickness changes visible over a sufficiently sampled single glacier?
- How do the findings compare to observed glaciological and geodetic glacier mass measurements?
- How does the reference DEM influence the quality of the results, and how to best model the footprint reference elevation?

## 2 Study site and data

### 2.1 Southern Norway

The study area referred to here as southern Norway extends over an area of 100'000km$^2$ at 59-63 degrees latitude. It comprises all areas of the Scandinavian Mountains south of Trondheim that are within a 20-km buffer around glaciers (Figure 1). While very steep especially at fjord flanks, the study area consists of both rounded and rough mountains but also includes high-elevation plateaus such as Hardangervidda. The climate of the study area is governed by a West-East gradient from a maritime climate at the coast with high precipitation amounts to dryer conditions further East in the rain shadow of the Scandinavian Mountains (Melvold and Skaugen, 2013). This is reflected also in measured glacier net balance magnitudes (Kjøllmoen et al., 2011). The Norwegian glacier area has recently been mapped by the Norwegian Water Resources and Energy Directorate (NVE) based on Landsat imagery from 1999-2006 (Winsvold et al. (2014); digital data available from the Global Land Ice Measurements from Space (GLIMS) database). Glaciers cover 1'522km$^2$ or roughly 1.5% of our study area. This includes 1'575 ice bodies ranging from small perennial ice patches of just over 0.01km$^2$ in size to the largest outlet glaciers (>40km$^2$) of the Jostedalsbreen ice cap. 50% of the glacierised surface in southern Norway consists of glaciers with <5km$^2$ spatial extension, and 20% of the glacier area of ice patches smaller than <1km$^2$. Some maritime glaciers advanced in the 90ties while glaciers located in more continental climate showed mainly frontal retreat (Nesje et al., 2008; Andreassen et al., 2005). After a culmination in 2000, most of the monitored glaciers in Norway experienced net mass deficit (Kjøllmoen et al., 2011; Andreassen et al., 2016).

### 2.2 ICESat

ICESat GLAS was a single-beam spaceborne laser altimeter operational between February 2003 and October 2009, sampling the surface elevation of the Earth within roughly 70m-footprints during two to three observation periods each year of about one month each (Schutz et al., 2005). The laser footprints have 172m spacing along-track, and approximately 42km cross-track spacing between 91-day repeat reference orbits at 61 degrees latitude (Figure 1). Cross-track spacing increases at lower latitudes, making polar areas in principle more favourable for ICESat applications. Note that our study area already lies in the polar acquisition mask of the ICESat mission at >59°N, where the off-nadir pointing mode enabled near repeats of the tracks (ca. +/-150m), in contrast to a nominal orbit repeat precision of +/-1'000m for mid-latitudes (Schutz et al, 2005). In

accordance with what Kääb et al. (2012) found to be the most suitable product for mountain glacier analyses, the ICESat data set used was GLAS/ICESat L2 Global Land Surface Altimetry HDF5 data (GLAH14), release 33 (Zwally et al., 2012). For GLAH14, elevation values were not changed between releases 33 and 34 (NSIDC, 2014). The data contains quality attributes and elevation corrections for each footprint. These attributes include a waveform saturation flag (attribute

*sat_corr_flag*) to indicate saturation of the sensor when recording the returned pulse, and a correction for the potential bias in extracted elevations from these saturated waveforms (attribute *d_satElevCorr*). The flags and corrections are intended for improving elevation accuracy on ice sheets, the original main purpose of the mission, and are not necessarily valid in rough mountain topography (NSIDC, 2012).

### 2.3 Reference data

The reference elevation datasets used are the national DEMs provided by the Norwegian Mapping Authority (further referred to as Kartverket) in 10m and 20m spatial resolution (http://data.kartverket.no). In mountain areas, the Kartverket DEMs are based on source data at 1:50'000 map scale including elevation contours at 20m equidistance, resulting in a nominal absolute vertical accuracy of +/-4-6 m (defined as the standard deviation of elevation; Kartverket, 2016). Using the source date stamp of elevation contours as a proxy, the age of the DEMs was found to be highly variable geographically, ranging from 1978 to

2009 on southern Norway's glaciers, and from 1961 to 2011 on non-glacierised areas.

For the Hardangervidda area and up to approximately 60.3°N, the global DEM from the Shuttle Radar Topography Mission (SRTM, Farr and Kobrick, 2000) is available at 3 arc-seconds resolution (corresponding to 93m in y, and 45m in x-direction at 60°N) from the U.S. Geological Survey (https://dds.cr.usgs.gov/srtm/). The SRTM DEM used here is based on C-band radar data acquired in February 2000 and consists of a composite of four or more overpasses at latitudes that far north (Farr

et al., 2007). The absolute vertical accuracy of the mission is stated as 16m (defined as 1.6 times the standard deviation of the error budget throughout the entire mission; Rabus et al., 2003) but found to be in the range of few metres as compared to ICESat elevations (Carabajal and Harding, 2006). The SRTM DEM featured as the reference DEM of choice for previous ICESat glacier trend analyses (e.g. Gardner et al., 2013; Kääb et al., 2012). Unfortunately, it does not cover glaciers visited by more than one ICESat overpass in southern Norway. In this study, SRTM version 3 serves as alternative reference DEM

for, thus, only land samples.

For parts of the non-glacierised Hardangervidda plateau, high resolution LiDAR DEMs were provided by NVE (Melvold and Skaugen, 2013). The data consist of six east-west oriented 80km long stripes of 500m width and cell size of 2m, flown on 21 September 2008 (minimum snow cover, leaf-off conditions). Datasets were available as high-resolution gridded DEMs. From comparison to a kinematic ground GPS survey carried out in April 2008, Melvold and Skaugen (2013) found

the absolute elevation errors of the LiDAR dataset to range from -0.95m to +0.51m, with a mean error of 0.012m and a standard deviation of 0.12m.

Yearly net surface mass balance estimates from in-situ measurements of 8 glaciers within the study area (see NVE's report series 'Glaciological investigations in Norway'; Kjøllmoen et al., 2011) were used as a reference for glacier behaviour during

the ICESat acquisition period. The data series are the product of the recent homogenisation of in-situ measurements with geodetic measurements (Andreassen et al., 2016) and are available from http://glacier.nve.no/viewer/CI (NVE, 2016).

## 3 Methods

ICESat data points from the end of the hydrological year (autumn campaigns) are treated as a statistical sample of glacier surface elevations in southern Norway. We follow the double differencing method described by Kääb et al. (2012) where differences between ICESat elevations and a reference DEM (hereafter referred to as dh) are analysed. Direct comparison of ICESat elevations of different years, as done for larger Arctic glaciers and ice caps (plane-fitting methods, e.g. Howat et al., 2008; Moholdt et al., 2010), is not possible for small mountain glaciers. These methods assume a constant slope of the ice surface within the spatial variability of ICESat repeat ground tracks, which is not given for small mountain glaciers. The use of a reference DEM instead takes into account the more complex surface topography of small glaciers. When compared to elevations from a reference data set of a different source date, the dh will be negative if the surface has lowered over time between the DEM source date and ICESat acquisition time, and positive if the surface has risen. Differences should be zero if the surface elevation was constant, such as over stable ground. Uncertainties in elevation measures of both datasets, not least as a result of rough terrain within the ~70m circular ICESat footprint, raise the need for sufficiently large statistical samples to reduce the effect of random errors. The evolution of dh over time is used to investigate surface elevation change trends over the ICESat acquisition period 2003–2008. (The 2009 autumn campaign is excluded due to low spatial coverage before complete ICESat failure.) Note that ICESat captures a signal of volumetric balance that results from surface elevation changes rather than mass change directly. The same is also the case where geodetic mass balances are obtained from DEM differencing, which is a widely used method. Comparison of ice surface elevation change trends with in-situ measurements provided in metres water equivalent (m w. eq.) requires unit conversion that depends on ice density. To validate the ICESat-derived trends, we back-converted the in-situ data using the same density as NVE used for mass/volume conversion of geodetic data (Andreassen et al., 2016), which is based on the findings of Huss (2013) who suggested a value of $850 \pm 60$ kg $m^{-3}$ as an average integrated over an entire glacier. (See also the discussion and density scenarios in the Supplement of Kääb et al., 2012.)

### 3.1 Pre-processing and filtering of ICESat data

ICESat surface elevations (height above reference ellipsoid) were converted to Norwegian height above mean sea level, in accordance with national DEM elevations. The ca. 170'000 data points within the study area were classified into *ice* and *land* footprints using the glacier outlines provided by NVE. Footprints lying partially on glaciers, i.e. with footprint centre locations within 40m of NVE glacier borders (both in- and outside original outlines), were classified as *ice border,* and excluded from further analysis. Apart from avoiding a mixed *land/ice* elevation signal from partly ice-covered 70m footprints this also accounts for the spatial uncertainty of glacier outlines and their potential change over time. For glacier

analyses, spring and summer campaigns were excluded to avoid biased trends due to yearly varying snow heights (see argumentation in Kääb et al., 2012; 2015), and the 2009 autumn campaign was excluded due to insufficient spatial coverage caused by weakening of the laser over time. To account for differences in spatial distribution and potential elevation changes due to onset of snowfall, the split autumn campaign of October 2008 (laser 3K, ran out of power before the campaign was completed) and December 2008 (laser 2D, completion of the autumn 2008 campaign) were treated separately where appropriate. *Land* footprints on fjords and lakes were filtered out using shoreline data provided by the Norwegian Mapping Authority, as water levels may vary (tides, hydropower reservoirs).

Reference DEMs were corrected for elevation bias and spatially co-registered with ICESat (see Sect. 3.2). Reference elevations for each footprint were extracted from the DEMs by different statistical means: footprint centre elevation, mean, median, mode (rounded to the metre/decimetre for the Kartverket/LiDAR DEMs), inverse distance-weighted (IDW, linear weighing, i.e. power 1), and bilinear interpolation of elevation of DEM grid cells within an assumed circular footprint with 35m radius (i.e., 4 grid cells for SRTM, 12 for Kartverket 20m, 38 for Kartverket 10m and ~960 for the LiDAR DEM).

The elevation differences between ICESat and the Kartverket DEM were analysed to denote a cut-off threshold for maximum elevation differences. Mean dh were found to be ~-0.5m for *land*, and ~-2m for *ice* samples (i.e. ICESat elevations are lower than reference elevations over glaciers). Using bootstrapping methods and histogram analysis for thresholds between 50m and 250m for |dh|, we found that a cut-off threshold of +/-100m dh effectively removed cloud measurements. Footprints with |dh|>threshold were excluded from all further analyses. The conservative threshold allows for uncertainty in elevation measurements of both datasets (*land* and *ice*), while allowing for slightly skewed dh distributions. It ensures all negative dh from glacier surface lowering between DEM acquisition date and ICESat elevation measurements are included while removing footprints on clouds (false positive dh).

Robust linear regression (we used Matlab's robustfit function with default parameterisation) through all individual samples was performed to find a linear trend for surface elevation change over time. Robust methods iteratively re-weigh least squares to find and exclude outliers until regression coefficients converge. For our *ice* trends we found that ca. 2-3% of the samples received weight 0 and were thus essentially removed as outliers. As an alternative trend estimate, we used the gamlss package in R ([www.gamlss.org](www.gamlss.org)) to perform regression using a fitted t-distribution. The t-fit accounts for the larger number of outliers in our distribution of dh (Figure 2) as compared to a normal distribution (Lange et al., 1989).

### 3.2 Sub-pixel shifts and corrections applied to the reference DEMs

Based on dh of autumn campaign *land* samples, elevation bias and spatial shifts between ICESat and the reference DEMs were quantified. The non-systematic spatial shifts of sub-pixel magnitude and biases were corrected, where possible. No corrections were applied to the LiDAR DEM. For the Kartverket and SRTM DEMs, directions and magnitudes of the shifts seemed to vary highly, also within single DEM tiles. Automated co-registration using the methods of Nuth and Kääb (2011) was performed to correct an overall 20 m south shift and -2.6 m vertical offset of the SRTM DEM, as compared to ICESat. However, additional shifts and biases that seem present in sub-units of the SRTM DEM could not be corrected. For the

Kartverket DEMs, dh were found to be elevation-dependent (more negative with increasing elevation above sea level *H*). The relationship is in the order of decimetres per 100 m elevation and applies to both the 10m and 20m DEM as both are based on the same source data. To account for this vertical bias, a correction term $c_H$ was applied to individual elevation values of both Kartverket DEMs:

$$c_H = 0.882 - 0.00158 * H \tag{1}$$

Automated co-registration of the individual nominal Kartverket DEM tiles (50x50km and 100x100km for the 10m/20m DEMs, respectively) was not applied systematically as it did not result in an overall positive effect. This is due to overlying shifts of (unknown) production sub-units within single tiles in different directions. To account for the apparently consistent vertical offsets in some areas, correction terms for each individual nominal tile ($c_{tile}$) and indicative source date ($c_{date}$) of the

Kartverket DEM were computed (after *cH* correction). For each nominal DEM tile the median *land* difference between ICESat and the Kartverket DEM was removed, or alternatively the same was done for each temporal unit of the Kartverket DEM. Both corrections are meant to remove vertical spatio-temporal biases and bias patterns in the reference DEM. The values of the corrections correspond to the median dh of all filtered *land* footprints at minimum snow cover (autumn campaigns only) per tile and date and are in the order of +/-1m per tile, and +/- 5m per date, respectively. Potential physical

causes such as vertical uplift due to post-glacial rebound in Scandinavia are in the order of decimetres for the last half century and cannot explain the large differences between ICESat and reference elevations on *land* surfaces. As a proxy for the reference DEM source date per ICESat footprint we used the time stamp of the closest elevation contour line to each footprint (elevation contours are the most important input dataset the Kartverket DEMs are based on; Kartverket, personal communication, 2013). However, these correction terms are approximate only as spatially confined units with unique source

data/firm update dates do not strictly exist and the total DEM is thus a product of spatio-temporal merging (Kartverket, personal communication, 2013), not untypical for DEMs from national mapping agencies.

For glaciers, spatially varying DEM source dates add additional uncertainty. Surface elevation difference between Kartverket DEM acquisition and the first ICESat acquisitions varies for individual glaciers, resulting in different (additional) offsets for each glacier. A correction term $c_{glac}$ for this effect was computed from the median dh of *ice* samples at the time of minimum

snow cover (autumn campaigns only) for each individual glacier, as classified using NVE's glacier inventory. The values of $c_{glac}$ range from -20m to +15m and reflect in this study mainly vertical glacier changes between the DEM and ICESat dates. For other areas potentially also other vertical biases from DEM production such as height datums or signal penetration could be addressed in a similar way. The latter are not relevant for the photogrammetric methods behind the Kartverket DEM, but for instance for radar wave penetration within the SRTM DEM.

**3.3 Sample representativeness and trend sensitivity**

In order to relate measured dh to actual net glacier mass balance, the ICESat sample has to mirror key characteristics of the area/terrain in respect to glacier driving processes. We assessed the representativeness of the ICESat glacier sample for the

study area in terms of average elevation, slope, aspect, spatial distribution of the footprints, glacier size, and age of the reference DEM. Representativeness in respect to terrain parameters was tested by comparing the sample distribution to the respective distributions of all glaciers in southern Norway (we used all Kartverket DEM cells within the glacier mask). This was done both for the entire ICESat sample and for individual campaigns. Consistency in terms of reference DEM age

distribution per campaign was assessed using the source date of the closest contour line for each sample as a proxy. Additionally, the size of the glaciers sampled by ICESat was compared to the entire glacier population of southern Norway.

To assure robustness of fitted glacier surface elevation difference trends, the effect of different data subsets and elevation corrections applied to either of the datasets were assessed. Subsets were created by including/excluding *a)* sets of footprints, as those classified as *ice border*, with specific DEM time stamps, or samples flagged as fully saturated (attribute

*sat_corr_flag* >=3), *b)* spatial subsets, e.g. of glaciers east and west of the main water divide, and *c)* entire campaigns. The elevation corrections assessed include ICESat saturation elevation correction (attribute *d_satElevCorr*) in addition to the correction terms per Kartverket DEM tile/source date/glacier described above ($c_{tile}$, $c_{date}$, $c_{glac}$). Very intentionally, we did not divide our sample into footprints only in the accumulation or ablation parts of the glaciers, respectively. In order to capture a signal that translates into geodetic mass balance it is essential to sample the entire glacier to consider both surface elevation

changes from ice melt/gain and dynamic glacier flow. If this is not ensured, the condition of mass continuity is violated, and it would thus be physically incorrect to draw conclusions on glacier mass balance based on surface elevation trends from a subset of samples in the ablation/accumulation areas only. The influence of separating footprints over ice and snow/firn for separate density scenarios is discussed in Kääb et al. (2012).

## 4 Results

### 4.1 ICESat sample overview

Roughly 75% of the nearly 170'000 ICESat footprints over southern Norway contain valid information of the Earth's surface elevation (125'312 samples after removal of footprints on clouds and water surfaces, see Table S1). Thereof, 2.6% lie fully on glaciers (versus an additional 0.9% that were classified as *ice border*). For glacier analyses, considering autumn campaigns only, a total of 1'268 *ice* and 48'854 *land* samples remain. These numbers are reduced by 2.8% (*ice*) and 1.6%

(*land*) only by excluding the weak autumn 2009 campaign. Dh of the remaining samples rarely exceed +/-10m. The dh are t-distributed with a narrower peak but heavier tails as compared to a normal distribution. Before application of the correction terms to the Kartverket reference DEM, the dh distributions of *ice* and *ice border* samples are considerably wider and in average more negative than *land* dh (Figure 2 left). After application of $c_H$, $c_{tile}$ and $c_{glac}$ correction terms, 94% and 95% of the *ice*, and *land* autumn samples, respectively, but only 80% of *ice border* autumn samples, show less than 10m absolute

elevation difference between ICESat and the (corrected) Kartverket 10m DEM elevations (Figure 2 right).

The spatial distribution and number of ICESat samples is not constant over time and decreases to as little as 10% of the number of samples of the autumn 2003 campaign, which includes most samples of all campaigns (427 *ice* samples). In

autumn 2009, only 35 ice samples (vs. 792 land samples) remain over southern Norway. Other autumn campaigns with very small sample numbers are 2005 (65 *ice* samples) and 2008 (24 and 24 *ice* samples for the October and December campaigns, respectively). These periods with particularly few samples correspond to campaigns with few orbits flown (2008, 2009) or heavy cloud coverage (2005).

128 of the *ice* samples lie on glaciers that were sampled only during one single autumn campaign. After the application of $c_{glac}$, any glacier elevation change signal from these single overpass samples is cancelled out. The majority of these (113) occurred during the autumn 2003 campaign due to the a transition between two different orbit patterns in the middle of that campaign (Schutz et al., 2005). The single overpass samples with, in average, 0m dh may thus flatten out derived trends and were excluded where appropriate.

**4.2 Representativeness of ICESat glacier sample**

The entire ICESat glacier sample appears representative in terms of elevation, aspect, slope, spatial distribution, and glacier area of the glaciers sampled (Figure 3, Fig. S1). Compared to the frequency histogram of the entire glacierised surface in southern Norway, ICESat slightly oversamples east-facing glaciers and underrepresents the glacierised area in the southwestern parts of the area of interest due to the orbits not covering the Folgefonna ice cap (Figure 1). However, these

deviations are of the same magnitude or less than deviations of the frequency histograms of the glacierised area monitored in-situ by NVE. Of the individual campaigns (autumn campaigns 2003–2008 shown within grey spread), those with fewest samples deviate most, but still follow the distribution of the full data set. Variability between campaigns is largest (wide grey spread) for easting, also for *land* samples, due to the sensitivity of the sample to exclusion of entire orbits (due to shorter campaigns / cloudy weather). The two autumn 2008 campaigns are only representative if combined as only a subset of orbits

was flown each in October and December, respectively. The autumn 2009 campaign was found to include *ice* samples only for one overpass (orbit 30, Figure 1), resulting in sampling of only Myklebustbreen and Haugabreen, an outlet glacier of the Jotunheimen ice cap. All other campaigns have 5-13 different orbits with glacier samples. Severe spatial concentration and poor representation of southern Norway's glaciers confirmed that also for our study area, the entire autumn 2009 campaign should be excluded from further analyses.

Of the 1'575 ice bodies in southern Norway, 96 or 6.1% are hit by at least one footprint of our filtered ICESat *ice* sample. While not the same glaciers are sampled each year, for all autumn campaigns except for 2009, footprints are spread on 17 (2008) to 77 (2003) different glaciers across the study area. Our ICESat footprints seem to capture small ice bodies according to their relative share of the total glacierised area: 47% of the samples lie on glaciers smaller than 5km$^2$, 17% on ice bodies <1km$^2$ (Figure 3 right). Only the (combined) autumn 2008 campaign samples no glacier >12km$^2$, and the ice

bodies sampled in December 2008 are distinctively smaller than those sampled in October in 2008. The smallest glacier within NVE's mass balance program in the area is 2.2km$^2$ large.

### 4.3 Error sources and corrections for ICESat and DEM elevations

Elevation errors in the DEMs were found to exceed ICESat footprint elevation uncertainty as well as the magnitude of corrections available in the ICESat products. ICESat elevation corrections from effects of waveform saturation (attribute *d_satElevCorr*) are in the range of decimetres; all other elevation corrections within the dataset are even smaller. Application of ICESat correction terms had no notable effect on dh distributions. The relative share of saturated samples (parameter *satCorrFlag* >=3 in the dataset) varies between 5-40% for the different campaigns, and is up to 15% higher for *ice* than for *land*. In contrast to the findings of Kääb et al. (2012) for High Mountain Asia, we found the number of saturated samples to decrease over time to as little 0-2% for the last three acquisition campaigns (laser 2D-2F). Filtering increased the relative share of saturated samples by on average 5%, and mean absolute dh (after filtering) are smaller for saturated footprints than for non-saturated ones (95% confidence) for both *land* and *ice*, whether or not saturation correction was applied to the dh. Saturated samples were therefore not removed from the dataset for trend computation, and saturation correction was not applied.

In contrast to the ICESat elevation values that seem robust without any corrections, elevation correction terms applied to the Kartverket reference DEMs significantly narrowed dh distributions (Figure 2 right). The elevation-dependent correction term $c_H$ successfully removed skewness towards more negative dh in dh-distributions, and per-glacier correction $c_{glac}$ clearly caused a major reduction in *ice* dh. The correction terms $c_{tile}$ and $c_{date}$ were found to be interchangeable and resulted in minor improvements only on *land* and *ice* dh distributions. For single footprints, uncertainty in reference DEM elevation is on the order of metres.

Looking at single footprints, reference DEM elevations differ by decimetres to metres between the different statistical measures (mean, bilinear interpolation etc.) applied to DEM grid cells within the ICESat footprint, for one and the same DEM. The method chosen matters most for the SRTM DEM with only four contributing cells, but differences resulting from the chosen elevation extraction method – from the perspective of a single footprint – are also higher for the high-resolution LiDAR DEM with ca. 960 contributing cells than for the 10/20m Kartverket DEMs. However, for larger sample numbers, these differences cancel out and dh distributions for reference elevations from the same DEM, but different elevation extraction methods, are approximately the same (Figure 4). Summarising statistical methods appear to produce slightly narrower dh distributions than centre DEM elevations only but the difference between the curves is not significant. Mode elevations differ most from reference elevations computed by the other methods, also for the 2m LiDAR DEM. We based our further analyses on median DEM elevations per footprint, or bilinear interpolation in the case of the low-resolution SRTM DEM.

Reference elevations between DEMs from different sources varied greatly. For the 184 autumn samples on Hardangervidda where all four reference DEMs were available, the LiDAR DEM matched ICESat elevations closest with a mean vertical offset of 0.03m and a narrow dh distribution (Figure 4). Elevation differences from the co-registered SRTM DEM are skewed with a heavier tail towards negative dh. Distributions of the (corrected) Kartverket DEMs, dating back to the 1970s

in eastern parts of the Hardangervidda, are particularly wide for this subset of samples, including an average vertical offset of -1.3m. For other spatial subsets, widths and vertical offsets of dh distributions of the SRTM and Kartverket DEMs vary to the same degree in a seemingly random way. Distributions of dh based on the 10m vs. 20m Kartverket DEMs were the same, also for other spatial subsets, and no improvement in elevation precision per footprint could be found from the finer grid

resolution.

Analysis of the DEM source dates for *ice* samples of the different campaigns (Figure 5) shows the representativeness of our sample in terms of Kartverket reference DEM age distribution. 70% of the samples have reference elevations from 2008–2009 (further termed 'post-2000'), and only approximately 20% and 10% date back to the 1990ies and 1980ies ('pre-2000'), respectively. Only two campaigns divert from this distribution: in autumn 2005, 60% of the ice samples have old reference

DEMs, and in 2009, all ice samples have very recent reference elevations from 2008–2009. For the split autumn 2008 campaign, all but one of the October samples fall on reference DEMs from 2008 while 80% of the December samples have pre-2000 DEMs. If using uncorrected Kartverket DEM elevations, pre-2000 dh are significantly more negative (mean dh: -7.3m) than post-2000 dh (-3.1m). The per-glacier correction $c_{glac}$ completely reconciles the two distributions as seen in Figure 2. Note that $c_{glac}$ treats glaciers as spatial units with consistent source dataset. Where this is not given – and parts of a

glacier surface are mapped on different dates or with different methods – the correction will be only partially effective.

## 4.4 Glacier thickness trends

We find a glacier surface elevation change of -0.39 ma$^{-1}$ +/- 0.07 standard error (1σ) for the years 2003–2008 (Figure 6 right) with all corrections to DEM elevations applied and samples on glaciers covered by only one single autumn overpass excluded. The trend slope decreases slightly to -0.34 +/- 0.062 ma$^{-1}$ when such single-overpass samples are included. Using a

t-fit instead, we found trends in general to be less sloping than robust trends for the same sample/set of applied corrections, and obtain alternative *ice* trend estimates of -0.33 +/- 0.07 ma$^{-1}$ and -0.27 +/- 0.061 ma$^{-1}$ on the same datasets. Campaign means are more negative than campaign medians, which indicates slightly skewed dh distributions for both *ice* and *land* samples. *Land* campaign means/medians follow the near-zero trend as computed from all individual samples very closely (0.05 +/- 0.009 ma$^{-1}$, t-fit: 0.04 -/+ 0.009 ma$^{-1}$). An exception to that is the December 2008 campaign which indicates surface

rise as compared to the October 2008 campaign due to onset of winter snow fall at higher elevations. Exclusion of the December 2008 campaign effectively sets the land trend to zero and renders the *ice* trend more negative. On the other hand, however, the December *ice* samples are required for the autumn 2008 campaign to be representative (see section 4.1). Correction of December samples for increasing snow depth (estimated from October-December *land* dh differences per elevation) also removes the *land* trend, but does not affect the *ice* trend. If the per-glacier dh correction $c_{glac}$ is not applied,

the *ice* trend is reduced and uncertainty increases to -0.26 +/- 0.12 ma$^{-1}$ (t-fit: -0.22 +/- 0.13 ma$^{-1}$). This decrease of thickness loss rate is due to the mixing of older and newer dates of the reference DEM that introduces biased dh and thus dilutes trends. Without the correction, *ice* campaign medians/means of uncorrected samples do not follow the assumed linear trend well and the standard errors of the campaign means just about overlap with 95% trend confidence bounds (Figure 6 left).

Deviation and uncertainty are largest for campaigns with few samples and non-representative DEM age distribution: 2005, (split) October/December 2008, and 2009 (excluded from trends). If ICESat trends were fitted through campaign medians instead of individual samples, these biased/non-representative campaigns would get the same weight as all other campaigns and, consequently, have more power to alter the derived trend. This stresses that ICESat trends over glaciers should be

computed based on the entire footprint sample, not based on campaign statistics (e.g. median dh) that give campaigns disproportionate weight compared to the actual number of samples included in that campaign.

After applying the per-glacier vertical correction $c_{glac}$ to the *ice* dh, means/medians of single campaigns follow the pattern of NVE's in-situ mass-balance measurements remarkably well. The range of cumulative net surface mass balances, converted to surface elevation changes(Huss, 2013), of eight glaciers in the study area is shown as grey spread in Figure 6 (Note that

this data is a product of the recent homogenisation of in-situ data of Norwegian glaciers with geodetic measurements (Andreassen et al., 2016) and thus differs from more positive glacier mass balance curves published earlier. For some of the studied glaciers, the data homogenisation suggests stronger mass loss and no or more moderate mass surplus for the glaciers with positive cumulative surface mass balance in the studied time period.) Campaign means are shifted up with the *ice* trend line crossing 0 m dh in autumn 2005 which corresponds to zero elevation difference between ICESat and reference DEM

considering decreasing sample numbers (autumn 2005 corresponds to the mean date of all ICESat samples used). Noteworthy is the 2005 autumn campaign which – only after correction – fits well with the reported positive net balance for five of ten measured glaciers (Kjøllmoen et al., 2006). The 2009 campaign does not follow the trend or the in-situ measurements, regardless of the application of $c_{glac}$. In-situ measurements suggest moderately negative net surface mass balances for that year (Kjøllmoen et al., 2010).

The slopes of both *land* and *ice* trends are not significantly affected ($< +/-0.01$ ma$^{-1}$ change in trend slope) by neither DEM correction terms ($c_H,$ $c_{tile}$ and $c_{date}$), the use of alternative statistical measures to extract DEM elevations per footprint, nor application of saturation correction to ICESat elevations. Exclusion of saturated samples and application of saturation correction to the remaining dh flattens out *ice* trend slopes by 0.03 ma$^{-1}$ and increases uncertainty (see Table 1). Including *ice*

*border* samples only affects the ice trend if $c_{glac}$ is not applied, but does not increase trend significance despite the increased sample number. If winter campaigns are included, the *ice* trend becomes considerably more negative (-0.43 +/- 0.066ma$^{-1}$, t-fit: -0.41 +/- 0.070 ma$^{-1}$). The same accounts for fitting a trend through winter campaign samples only (-0.42 +/- 0.092, t-fit: -0.41 +/- 0.097 ma$^{-1}$). Note that for comparability between winter and autumn trends single overpass samples are not excluded in the numbers here. The 2003 winter campaign had a different orbit pattern than later campaigns (Schutz et al., 2005). We

found yearly varying snow heights of between 3 to 7m on glaciers, and the maximum values in winter 2005 correspond well to the overall strongly positive winter mass balance of that particular year (Kjøllmoen et al., 2006). *Ice* trend slopes are considerably more sensitive to all changes in sample composition described above if $c_{glac}$ is not applied.

Continental glaciers east of the water divide show a more negative trend than coastal glaciers. The same is true for small (area <5km$^2$) versus large glaciers, and *ice* samples with pre-2000 vs. post-2000 reference DEM. The latter corresponds to an arbitrary subset in size (with a tendency of older reference DEMs for smaller glaciers) and spatial distribution of glaciers rather than a selection based on any physically meaningful criteria. The increases in trend slope amount to 50-150% between these respective subset pairs (Table 1). However, we could not find a significant relationship between dh magnitude and distance to coast. Exclusion/inclusion of entire campaigns was found to affect trends only for campaigns at either end of the ICESat acquisition period.

Note that also subsets of samples of only accumulation/ablation zones, as well as certain elevation or slope classes, would result in different trends (not shown). Such sample subsets can obviously not fulfil the requirement of representativeness for the entire glacier area and are thus not comparable to in-situ glacier mass balance measurements. Glaciers that are not in balance adjust their geometry via glacier flow which causes additional surface elevation changes that may be different for the accumulation and ablation parts of a glacier. Only sampling of the entire glacier(s) ensure that both elevation changes due to surface mass balance as well as glacier dynamics are included in the volumetric mass balance signal measured by ICESat.

The problem of biased trends due to non-representative spatial sampling by ICESat is illustrated well by the spatially clumped autumn 2009 campaign. The only glaciers that are sampled in 2009 have a strongly positive trend (Figure 7, +0.47 +/-011.ma$^{-1}$, in total 181 samples from Myklebustbreen and Haugabreen for autumn campaigns 2003–2009). While this trend is based on fewer campaigns (missing data in 2005 and 2007, only 3 and 7 samples for the 2004/2008 campaigns, respectively), the trend slope is not unrealistic (2.05/0.14 m w.eq. cumulative balance before/after data homogenisation for nearby Nigardsbreen in 2003–2009; Kjøllmoen et al., 2009; Andreassen et al., 2016). The ICESat sample on these glaciers is representative (also for single campaigns) in terms of elevation, slope, aspect and spatial distribution (within a single track that roughly follows the glacier flowline) as compared to the entire glacier area of Myklebustbreen/Haugabreen from the reference DEM. The reference DEM for this area was updated in 2008, resulting in a positive offset of the *ice* campaign mean in autumn 2009 (Figure 6).The fact that these glaciers are not at all representative for the cumulative mass balance of the entire glacier population in southern Norway explains the large offset of the 2009 campaign mean to the 2003–2008 ICESat trend.

## 5 Discussion

### 5.1 Representativeness

When combined with reference elevations from a DEM, ICESat data provides realistic estimates for glacier surface elevation change in southern Norway. However, our results bring out the importance of ensuring representativeness of the sample as well as good control over biases in reference elevations.

The ICESat sample has to be representative not only in terms of terrain and topographic characteristics that govern glacier behaviour but also data quality aspects that vary spatially. Parameters with coarse spatial patterns have largest biasing potential. Consequently, reference DEM quality and age, glacier area, and severe variations in spatial distribution of the samples were found to have potentially largest impact on glacier trend estimates. This sensitivity is a direct result from interference of the non-uniform glacier behaviour within the study area with the (coarse) spatial pattern of these influencing parameters. In contrast, parameters that vary much more spatially such as elevation, slope or aspect were found to be of less concern. Also smaller sample subsets are representative in that respect. Campaigns with low sample numbers and spatial clumping are most prone to biases. Owing to the rapidly decreasing laser power, mostly campaigns towards the end of the acquisition period are affected. However, severe cloud cover and subsequent exclusion of too many orbits can result in poor spatial distribution also for other campaigns. An example for this is the autumn 2005 campaign in southern Norway for which the only few *ice* samples mostly lie on old reference DEMs.

When relating ICESat trends to traditional glaciological measurements it is important to keep in mind that the subset of in-situ monitored glaciers and the glaciers covered by our ICESat sample might not be fully comparable. Differences in estimated mass/volume changes are therefore likely not (only) caused by the methods used, but rather a result of different sample composition. This is in line with the findings of e.g. Zemp et al. (2015) or Cogley (2009) who assign differences in mass budgets as from glaciological vs. geodetic measurements to sample composition rather than method-inherent causes. We find that with ICESat's random spatial sampling (with respect to glacier locations), we capture also many small ice bodies and snow patches. The share of samples, in terms of the area of the ice bodies where single footprints lie on, accurately reflects the size distribution of all glaciers and ice patches of the total glacierised surface in southern Norway. While such small ice patches are commonly not monitored in-situ, they are likely to be equally affected by climate change if not even more sensitive (Bahr and Radić, 2012; Fischer et al., 2014). Subsequent differences in glacier volume/mass changes as derived from ICESat, compared to traditional glaciological methods on selected valley glaciers, might therefore not agree if upscaled to the entire glacier population of a study area (Bahr and Radić, 2012).

The moderately negative glacier surface elevation change trends for the years 2003–2008 fit well with overall negative net cumulative mass balance series from glaciological measurements on glaciers in southern Norway. Trend slopes are robust against applied corrections or changes in sample composition as long as representativeness of the sample is guaranteed. Given the highly heterogeneous behaviour of Norway's glaciers and the varying age of some parts of the reference DEM, both the measured dh (up to 20m) and the resulting trend confidence intervals are within an expected range. We find that smaller glaciers, and glaciers to the (dryer) east of the water divide, experienced stronger changes than larger and coastal glaciers. This is in agreement with the individual reactions of the monitored glaciers in southern Norway to the increasing atmospheric temperatures during the last decade.

To fill gaps from missing campaigns, or to increase spatial resolution of estimated glacier trends, other authors have tried to obtain an alternative trend estimate fitted through winter *ice* samples (e.g. Gardner et al., 2013). However, our results for southern Norway show that ICESat is sensitive to – and even able to reproduce – yearly varying snow depths, and our glacier surface elevation change trends are more negative for winter *ice* samples. Even though the difference between the winter and autumn trends is not significant in our study, the standard error of the winter trend is 50% larger which reflects the uncertainty added from yearly/spatially varying snow depths. Moreover, the different orbit pattern of the winter 2003 campaign (and first phase of autumn 2003 campaign) as compared to all following campaigns may cause problems with representativeness and spatial distribution of the samples, especially if spatially varying elevation corrections such as our per-glacier correction $c_{glac}$ are applied. Our results therefore advise against including winter samples in glacier trend analyses. We also recommend including only footprints lying entirely on glaciers, i.e. excluding footprints that we classified as *ice border* samples. The signal from mixed ice/land footprints adds unnecessary uncertainty to the derived trends that does not justify the increased sample numbers.

On the example of Myklebustbreen, we show that it may be possible to detect trends even for single glaciers. Unfortunately, no mass balance measurements exist to verify the positive surface elevation change found for this glacier. How confident we can be in such a local trend depends on appropriate temporal and spatial coverage. Our results show that the applicability of ICESat in arbitrary glacierised regions does not depend on a single factor only. Likewise, the minimum region size needed to derive valid estimates on glacier surface elevation change from ICESat cannot be expressed as a hard threshold but depends on a combination of factors specific to each area: Glacier density and ICESat track density (i.e. sample size), representativeness of the ICESat sample, and homogeneity of the glacier signal within the study (sub-) region. In general, ICESat track density increases with latitude, making areas closer to the poles more favourable for ICESat studies. However, size and spatial distribution of glaciers as well as less cloud cover in dryer areas may result in large enough sample numbers even in small mountain regions at lower latitudes – as long as the representativeness condition is fulfilled. Representativeness of the sample may be given also for lower sample numbers than we found in southern Norway where a glacier population is more homogeneous in respect to its topographic setting as well as mass balance changes/surface elevation trends. Spatially varying effects such as from DEM elevation bias or highly non-uniform glacier behaviour within the study area require larger sample numbers – and thus larger region sizes – to account for the introduced uncertainty. In that regard, southern Norway may not be an ideal location to test the limits of ICESat applicability, and in other mountain regions with more consistent reference DEMs even smaller study areas may potentially yield valid ICESat glacier surface elevation change estimates.

**5.2 Glacier trend sensitivity**

Given the temporal variability in annual surface mass balances from NVE's long-term measurements, the glacier surface elevation change derived from ICESat data is not likely to represent a long-term trend. Our results are only representative for

the development within the five years covered. It is in general not recommended to extrapolate trends derived from such a short time interval, neither for ICESat-derived trends, nor mass-balance series in general.

Trend slopes are considerably less sensitive to missing/biasing campaigns in the middle of the ICESat acquisition period than to campaigns missing at either end. Inclusion of the non-representative 2009 campaign which diverges strongly from the assumed linear trend (corresponding to an assumed constant mass balance) significantly alters the trend slope. The considerable trend slope differences for our various sample subsets show that trends are even more sensitive to changes in sample composition or applied corrections when sample numbers are small. .

For our data, we found that robust fitting methods, as used by e.g. Kääb et al. (2012) for ICESat glacier trends, result in comparable but somewhat steeper trend estimates as when fitting a t-distribution to the data. The error estimates of both methods overlap for all subsets/sets of corrections applied to the dataset, thus the trends are not significantly different. A t-fit better captures the heavier tails of the sample distribution and includes the uncertainties in the data within the statistical model used to compute the fit. The iteratively lowered weighing of samples within the robust fitting technique (which assumes a normal distribution) results in a similar effect – although one can argue that the weights assigned to outliers are so small that data points that don't fit the trend essentially are removed, and thus sample numbers reduced. Consequently, according to Street et al. (1988), error estimates for the robust methods might not be correct. However, given that most outliers indeed correspond to erroneous measurement of either ICESat or reference elevations, exclusion of these samples from trend estimates might be desirable. We found that error estimates of both methods are very similar, and differences resulting from the different trend fitting approaches are of the same order as caused by changes to the sample composition or due to application of correction factors. We thus prefer to leave open if robust or t-fits are more appropriate to derive elevation trends from ICESat.

## 5.3 The role of DEM quality and elevation errors

Of all correction factors applied, the correction for constant offsets on glaciers introduced by DEM age ($c_{glac}$) deserves special attention as it considerably increased the statistical significance of glacier surface elevation trends. Not only is the trend standard error halved, but the correction also makes the trend slope much more robust to changes in sample composition/elevation corrections applied. The correction thus captures and eliminates errors in the dataset that have a far bigger effect on trends than for example different fitting techniques. By applying $c_{glac}$ we see an increase in trend slope even though the correction decreases *ice* dh. The fact that also single campaigns fit measured mass balance after application of the correction strongly indicates that this correction is important to accurately capture glacier surface elevation development within the studied time period. The estimated glacier surface elevation trend of the sample without accounting for DEM age offsets is not significantly different from the former trend estimate, but the wider confidence interval, trend sensitivity, and large offsets of single campaigns, are a clear sign that not all error sources were accounted for in the uncorrected dataset. It also illustrates very well the importance of representativeness in terms of maybe not immediately obvious factors such as

spatially varying vertical offsets in the reference data. Note that a correction for "DEM age", as we do it here, has a different importance on glaciers as compared to stable ground. On glaciers that change their surface elevation over time the spatially varying bias we see in our dataset is likely indeed caused by different DEM ages. On top of that other spatially varying biases due to mosaicking of data from different sources may add additional bias on glaciers. On land surfaces, the contrary is the case and the latter type of bias would usually play the main role – while the age of the reference DEM is negligible except for areas and timescales where e.g. vertical uplift due to post-glacial rebound causes relevant age-dependent bias.

Where the correction is applied on spatial units with changing elevation – such as on glaciers – a certain consistency and repetition in spatial sampling is needed. The surface change signal contribution from a glacier sampled only by one overpass is removed by the $c_{glac}$ correction. While we found that the error from keeping the single overpass samples in our trend estimates is smaller than the uncertainty from not applying $c_{glac}$ we recommend removing these samples as the introduced bias corresponds to a systematic flattening of the trend. It should be kept in mind that for winter trends (summer trends on the southern hemisphere) this might affect most, if not all, of the March 2003 campaign samples due to the different ground track pattern of that campaign.

Correction of per-glacier offset is only possible in our study because the glaciers seem to mostly correspond to spatial units of consistent DEM age in Norway. The correction factor is independent of (not available) metadata for data quality and does not correspond to nor help to correct offsets of the surrounding terrain. In our case, zero *land* trend therefore does not guarantee the absence of a time-dependent bias for glacier samples (with different distribution in terms of source date stamp). The assumption of a constant vertical offset per glacier is not necessarily valid everywhere – e.g. Swiss glaciers were not considered as unities in the mosaicking of airborne DEM acquisition flight lines but sometimes cut right across (Martin Hölzle, personal communication, 2015). This resulted in differently timed outlines and elevation data for parts of the same glacier, further complicating DEM differencing with historic DEMs in the Alps, as done by Fischer et al. (2015). We faced similar challenges in our attempts to co-register ICESat and the reference DEMs. The spatial units (tiles or source time stamp of elevation contours) available to us did not correspond entirely with spatial units of data origin that would exhibit a constant spatial shift or elevation error. Other DEMs for larger areas, and especially national DEMs, are likely to contain similar inherent errors as we found for the Kartverket DEM, and Fischer et al. (2015) for historic Swiss DEMs, as they all consist of a patchwork of source datasets with various time stamps – especially in remote areas. Metadata on elevation data sources are rarely available, and DEMs might have been (post-) processed to optimize characteristics other than high elevation accuracy, for instance smoothness or realistic visual appearance.

Also global DEMs, for instance the ASTER GDEMs or the upcoming TanDEM-X DEM, might be a composite of numerous units of unknown or different age or elevation biases. While the radar-based elevations from the SRTM were acquired within a short time frame which eliminates DEM age error, the DEM still remains a patchwork from acquisitions from different overpasses, and elevation differences to ICESat elevations were found to vary spatially (e.g. Carabajal and Harding, 2006). Van Niel et al. (2008) found that shifts of sub-pixel magnitude result in artificially generated elevation differences of the same magnitude as the actual, measured elevation differences between the SRTM and national higher-resolution DEMs for

two mountainous test sites in Australia and China. As an additional source of uncertainty for radar-based DEMs when serving as reference elevation, radar penetration into snow and ice is estimated to be in the range of several metres (Gardelle et al., 2012; Kääb et al., 2015) and can be considered another type of spatial pattern where our per-glacier correction could be of benefit. However, further analyses on this end would be necessary, given the strong gradients and differences in

snow/ice consistency between accumulation and ablation zones of a glacier that make radar penetration vary strongly even within a single glacier (Dall et al., 2001; Müller, 2011; Rignot et al., 2001).

ICESat GLAS data comes with numerous correction terms which might signal uncertainty in the elevation values. On the example of saturation correction, which is in the order of decimetres, we showed that the effect of these corrections is

negligible over rough mountain terrain and not affecting our results. Moreover, the saturation flag does not necessarily correspond with lower quality data over mountainous terrain, also not on ice surfaces in the mountains. The correction might not capture the effect of waveform saturation over such terrain appropriately. It is not generally recommended for land surfaces (NSIDC, 2012), and the error potentially resulting from waveform saturation is in the order of decimetres only. However, Molijn et al. (2011) found a larger occurrence of saturated samples at the transition from (rough) glacier-free

terrain to (flat) glacier surfaces in the Dry Valleys in Antarctica. This can be explained with the adaptive gain setting of ICESat's GLAS instrument: The gain of the sensor is dynamically adjusted based on the recorded signal (NSIDC, 2012) and might not adapt fast enough for an abrupt change in the recorded waveform shapes between a footprint on dark, rough rocks and a flat, bright ice surface. A preferred occurrence of saturated samples and subsequent elevation error at glacier margins, where surface elevation changes are likely more pronounced, could potentially lead to a systematic bias in ICESat-derived

glacier surface elevation change trends. In our study area we could not detect a systematic pattern in the spatial distribution of the saturated samples, also not where ICESat is passing over glacier margins and experiences a land/ice surface type change. We believe that this is due to the small size of mountain glaciers and the rough surface topography both on land and glaciers (as compared to large Antarctic outlet glaciers) that never really allow the sensor to settle for a certain gain. Nevertheless, from the findings of Molijn et al. (2011) we cannot exclude that there is a potential for a systematic bias from

waveform saturation at ice/land transitions in other areas, and we recommend to consider this possibility when applying our method in an arbitrary glacier region.

Likewise, other available corrections and biases of even smaller magnitude, such as inter-campaign bias (<8cm, Hofton et al., 2013), the optional range increment for land samples (*d_ldRngOff*), and the GmC correction introduced in GLAS data of release 34, are of negligible importance compared to corrections applied to the reference DEM elevations. However, it

cannot be excluded that these corrections might become relevant if a reference DEM without vertical bias were available – which would eliminate the current main error source.

On stable ground, the problem of time-dependent elevation differences due to surface elevation change is not present, but the artificial dh resulting from sub-pixel shifts or elevation-dependent errors were still found to compete with real, measured

differences between the DEMs. This mainly has implications on the size of spatial and temporal units needed to aggregate footprints to get meaningful results. The example of Hardangervidda illustrates the potential of results on a local scale for areas with good quality reference elevations. Thereby, spatial resolution of the reference DEM is of less importance than the absence of (spatially varying) shifts or other biases in the data, resulting in narrower dh distributions of the low resolution

SRTM DEM as compared to the Kartverket DEM which seems to be of poorer quality in this area. However, the DEM resolution has to be small enough to appropriately capture the local relief variations. In more rugged terrain with large elevation variation within a single footprint, the spatial resolution of the DEM would likely play a more important role than on rather flat areas like Hardangervidda. We found the reference DEM rather than ICESat to limit e.g. more localised results that would reflect spatial variation or patterns of glacier change within the study area.

For glacier trend applications, the time to collect better reference DEMs for improved retrospect ICESat analyses has likely passed where glaciers experienced large changes in volume over the past decade. Still, the biases in the old reference DEMs of our study, originating from 10 to 20 years prior to the ICESat acquisition period, obviously became detectable and quantifiable. This fact underlines that ICESat data fully bears the potential to serve as a sample of glacier surface elevations in the 2000s even for areas where we currently do not yet have very accurate reference DEMs.

**6 Conclusion**

On the example of southern Norway, we show that ICESat elevations normalised to a reference DEM are fully capable to provide robust and realistic glacier surface elevation trends for the years 2003–2008 in mountainous terrain with scattered small and medium size glaciers. We estimate an average ice surface elevation change of -0.39m +/-0.07 (robust fit) and -0.33m +/-0.07m (t-fit) ice per year in 2003–2008 for southern Norway's glaciers. Our estimate corresponds very well to the

area-weighted average of observed cumulative mass balances from in-situ and geodetical mass balance measurements on 8 glaciers in the study area.

Despite sparse glacier cover of the study area, the coarse spatial sample of ICESat represents southern Norway's glaciers accurately in terms of elevation, slope, aspect, spatial location, and area of the glaciers. Representativeness of the sample is given also for individual campaigns which is a prerequisite for robust trend results. Non-representative campaigns have the

potential to alter trends. Especially in terms of glacier area, ICESat samples reflect the size distribution of all glaciers in southern Norway considerably better than the (predominantly large) glaciers included in the in-situ mass balance network in Norway.

The number of ICESat footprints on glaciers (1'233 after filtering) within the study area was found large enough to allow for spatial and thematic subsampling. The considerable differences between trends from different sample subsets reflect the

wide range of observed cumulative mass balances in the study area. Reasonably, we see a more negative elevation trend of continental and small glaciers as compared to coastal or large glaciers, respectively. Our glacier elevation change trends thus capture very varied glacier behaviour within the study area, and depict also glaciers with positive mass balance, as seen for

Myklebustbreen and Hansebreen. On this example, we show that it may be possible to detect trends even for single well-covered glaciers, however with increased uncertainty due to spatially clumped sampling and missing data for some campaigns.

The applicability of ICESat in arbitrary glacierised regions depends on a combination of factors rather than a minimum region or sample size. The number of samples is determined by glacier density in relation to ICESat track density and the topography/climate-determined fraction of valid elevation measurements in the study region. Their representativeness, however, depends on the homogeneity of both the glacier topographic setting and their mass balance signal within the study area, as well as other spatially varying effects such as from DEM elevation bias. These factors are inherent for each region (and reference DEM) and will affect the sample/area size needed for a valid surface elevation change estimate. Uncertainties in reference DEM elevations exceed ICESat uncertainties by a magnitude. Elevation bias of unknown spatial units of the three assessed reference DEMs add noise that match, or exceed, measured elevation differences. These biases result from sub-pixel horizontal and vertical shifts, elevation-dependent bias, and varying source time stamps of the reference DEM of up to 20 years prior to ICESat acquisition. If not accounted for, spatially varying biases in combination with varying sample distribution over time may not cancel out, and can affect the results by causing false trends. Representativeness of the sample in terms of such spatially varying bias in the reference DEM was found to be more important (and less given) than for terrain parameters like elevation or aspect. Due to their coarse spatial pattern, the DEM errors add varying but systematic bias – in contrast to the random effects from geographic ICESat footprint distribution.

We developed a new per-glacier correction to harmonise the effect of age-dependent offsets between ICESat and the patchy reference DEM of unknown, but spatially varying source date. This correction greatly increased the statistical significance and robustness of our glacier change trend, and also single campaigns fit measured mass balance after application of the correction. For national or global DEMs in other regions, we see large potential from this correction, or modified versions of it, for reducing glacier trend uncertainty related to spatio-temporal biases such as from imperfect mosaicking, orbit inaccuracies, or radar penetration.

Our study shows that ICESat analyses in mountain terrain currently are limited by the reference DEMs rather than ICESat performance. ICESat provides an accurate sample of global glacier surface elevations in the 2000s. There is still large potential, even several years after the mission ended, that new upcoming DEMs could improve ICESat analysis in retrospect (e.g. TanDEM-X, new mapping agency DEMs). After its launch, ICESat2 with its denser cross- and along-track sampling and improved performance over rough surfaces (Kramer, 2015) will have the capability to provide even more detailed and accurate valuable sample of glacier surface elevations using the methods outlined here.

## Author Contributions

D. Treichler designed the study, performed data analyses and prepared the manuscript. A. Kääb designed the study and edited the manuscript.

**Acknowledgements**

The study was funded by the European Research Council under the European Union's Seventh Framework Programme (FP/2007–2013)/ERC grant agreement no. 320816, the ESA project Glaciers_cci (4000109873/14/I-NB) and the Department of Geosciences, University of Oslo. We are very grateful to NASA and USGS for free provision of the ICESat data and the SRTM DEM used, respectively, to the Norwegian mapping agency for their topographic DEMs, and to the Norwegian Water Resources and Energy Directorate for glacier outlines, mass balance data, and the Hardangervidda LiDAR DEM.

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

**Table 1.** Trends and trend standard error (se), as computed from different subsets and corrections applied to the dataset ($c_H$, $c_{tile}$ and $c_{glac}$ are applied unless specified otherwise). Footprints on glaciers sampled only during one autumn campaign are excluded except for the subsets marked with an asterisk, i.e. * corresponds to all 2003–2008 (autumn) *ice* samples.

| Dataset | Correction / subset | robust trend | se (1σ) | samples | t-trend | se (1σ) |
|---|---|---|---|---|---|---|
| *ice* | *($c_H$, $c_{glac}$, only > 1 overpass)* | **-0.39** | **0.07** | **1'105** | **-0.33** | **0.07** |
| *land* | *($c_H$, $c_{tile}/c_{date}$)* | *+0.05* | *0.009* | *48'089* | *+0.04* | *0.009* |
| *ice* | *($c_H$, $c_{glac}$)* all *ice* samples * | **-0.34** | **0.062** | 1'233 | **-0.27** | **0.061** |
| *ice* | $c_{glac}$ not applied * | -0.26 | 0.12 | 1'233 | -0.22 | 0.13 |
| *ice* | Dec 2008 excluded | -0.44 | 0.072 | 1085 | -0.37 | 0.071 |
| *land* | *Dec 2008 excluded* | *-0.003* | *0.010* | *44'568* | *-0.004* | *0.010* |
| *ice* | Corr Dec 2008 | -0.4 | 0.07 | 1105 | -0.34 | 0.069 |
| *land* | *Corr Dec 2008* | *+0.001* | *0.009* | *48'089* | *-0.003* | *0.009* |
| *ice* | Incl 2009 | -0.25 | 0.065 | 1'140 | -0.22 | 0.066 |
| *land* | *Incl 2009* | *+0.03* | *0.008* | *48'854* | *+0.03* | *0.008* |
| *ice* | *Sat_corr* applied, saturated samples excluded | -0.35 | 0.072 | 1'001 | -0.3 | 0.075 |
| *ice* | East of water divide | -0.55 | 0.14 | 242 | -0.54 | 0.14 |
| *ice* | West of water divide | -0.36 | 0.08 | 863 | -0.29 | 0.08 |
| *ice* | Pre-2000 DEM source date | -0.72 | 0.16 | 298 | -0.64 | 0.17 |
| *ice* | Post-2000 DEM source date | -0.29 | 0.076 | 807 | -0.26 | 0.076 |
| *ice* | Including ice border samples | -0.36 | 0.07 | 1'541 | -0.33 | 0.07 |
| *ice* | Including winters 03–08 * | -0.43 | 0.066 | 2'536 | -0.41 | 0.070 |
| *ice* | Only winters 03–08 * | -0.42 | 0.092 | 1'303 | -0.41 | 0.097 |
| *ice* | Samples on glaciers > 5 km$^2$ | -0.28 | 0.089 | 621 | -0.26 | 0.091 |
| *ice* | Samples on glaciers < 5 km$^2$ | -0.53 | 0.11 | 484 | -0.43 | 0.11 |
| *ice* | Myklebustbreen/Haugabreen (03–09) | +0.47 | 0.11 | 181 | +0.47 | 0.12 |

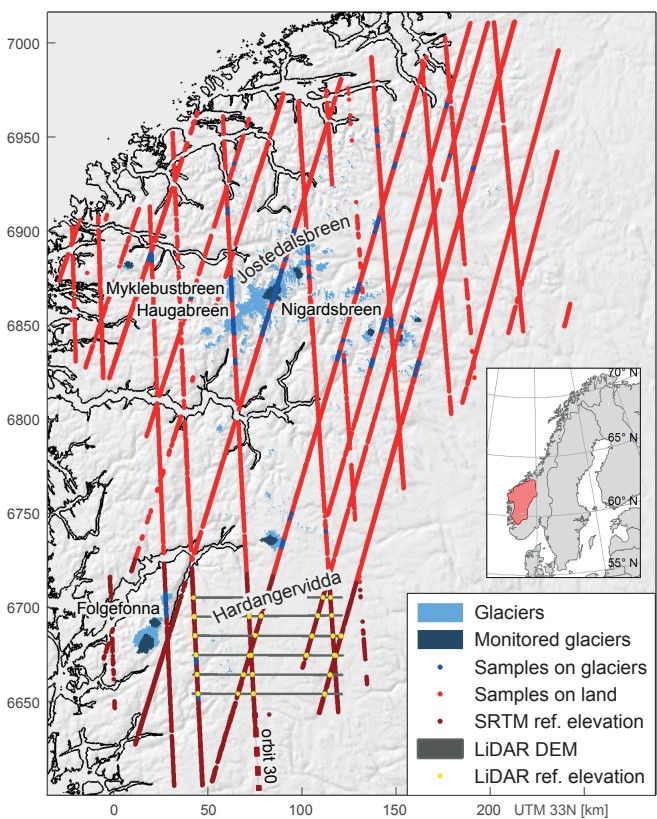

**Figure 1**. ICESat samples over glaciers and stable ground (land) in southern Norway. Only used footprints are displayed (no footprints on clouds or water). Glaciers with on-going monitoring by NVE are emphasised.

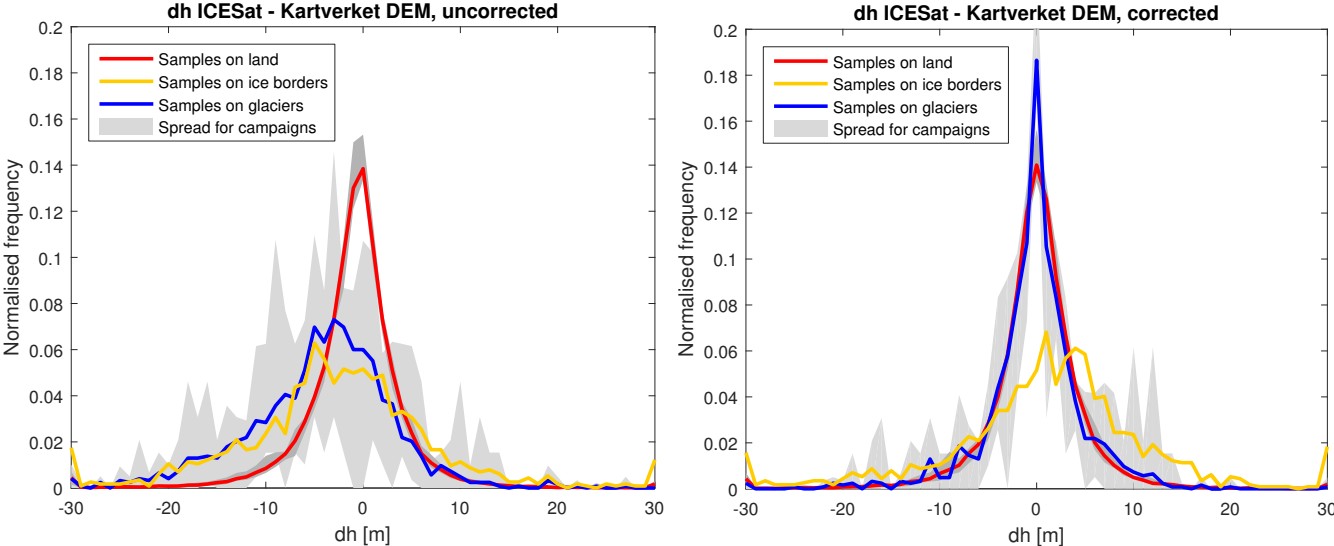

**Figure 2.** dh of *land* and *ice* (autumn campaigns 2003–2008) for the uncorrected Kartverket 10m DEM elevations (left) and with DEM elevation corrections ($c_{tile}$, $c_H$) and per-glacier correction ($c_{glac}$) applied (right). The grey spreads shows the range of distributions for *ice* (wide spread, light grey) and *land* dh (narrow spread, darker grey) of individual campaigns.

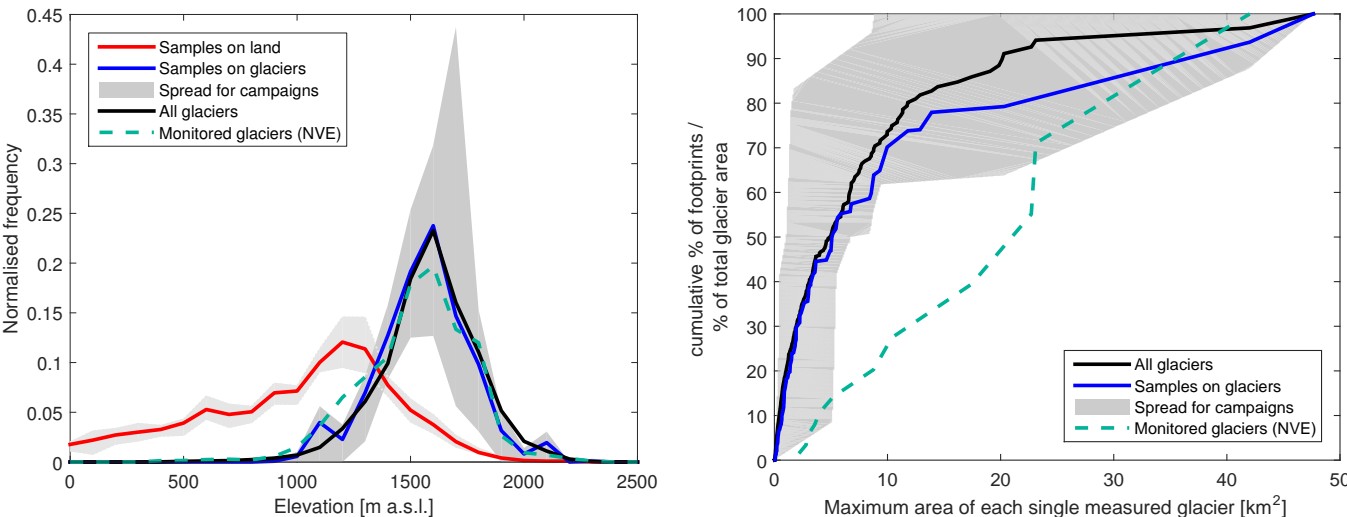

**Figure 3.** Representativeness of 2003–2008 ICESat autumn campaign samples in terms of footprint elevation (left) and area of glaciers sampled (right), compared to the entire glacierised surface in southern Norway, and to monitored glacierised surface (mass balance program by NVE). The grey spread encompasses the distributions of single ICESat autumn campaigns; where it is wide, the difference between individual campaigns is largest. (Reading example for glacier area comparison: 50 % of the entire glacierised surface in southern Norway is made out of glaciers < 5 km$^2$, 50 % of the glacierised surface where NVE runs a mass balance program is made of glaciers < 23 km$^2$, and 50% of all ICESat autumn ice samples lie on glaciers < 5.1 km$^2$.)

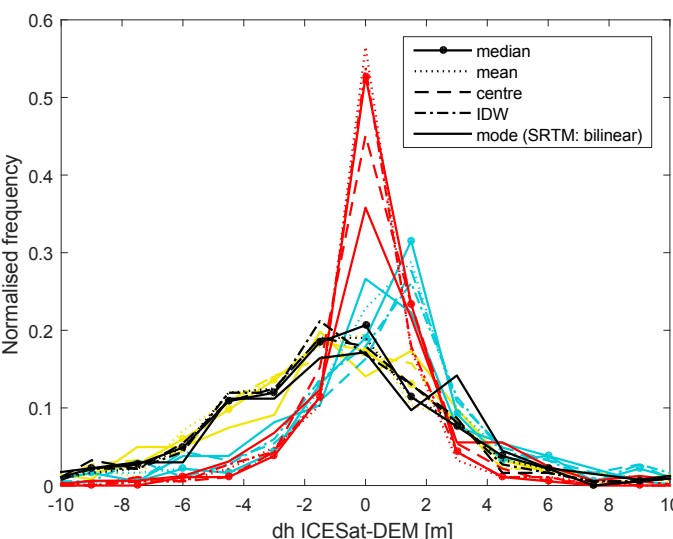

**Figure 4.** dh from different reference DEMs and statistical measures to summarise elevations within footprints (184 *land* samples): LiDAR 2m (red), Kartverket 10m (black) and 20m (yellow), SRTM ~90m (blue, bilinear interpolation shown instead of mode).

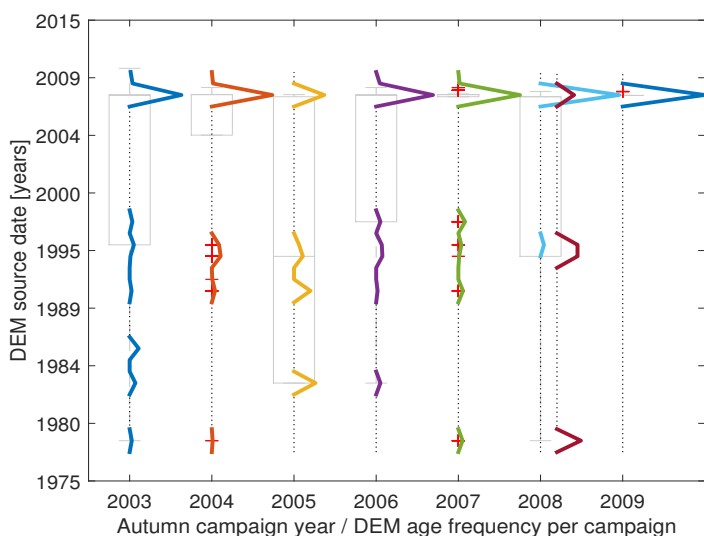

**Figure 5.** DEM source date distributions for ICESat autumn campaign samples on glaciers. The boxplots emphasise the average DEM age per campaign while the frequency histograms (coloured curves) reflect the relative DEM age distributions. In 2008, the October (blue) and December (brown) campaigns are shown separately (frequency histogram) and grouped (boxplot).

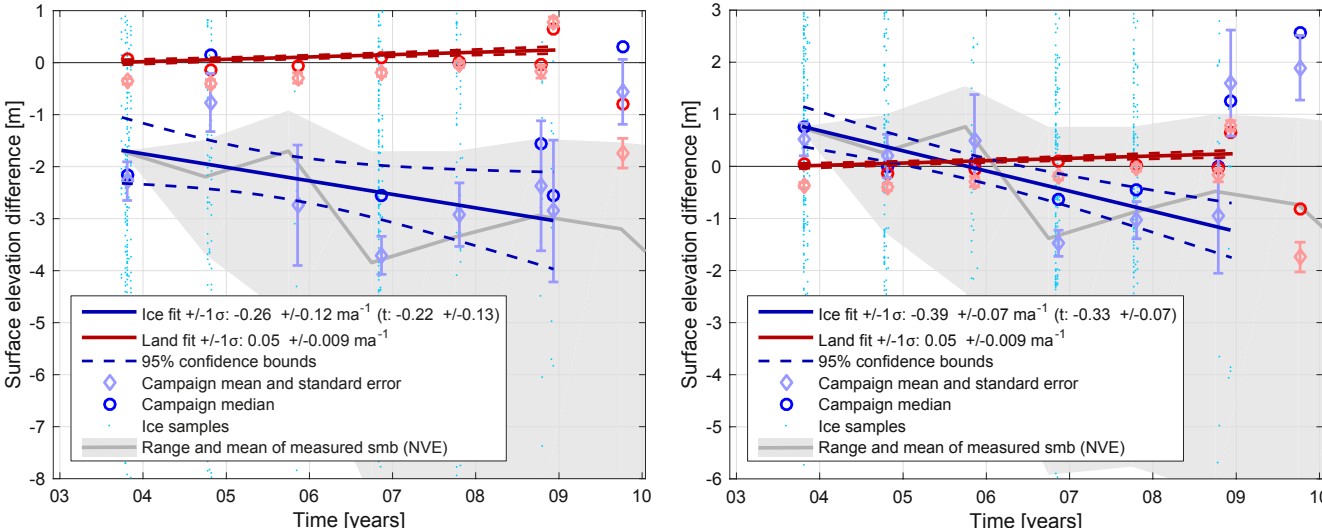

**Figure 6**. Surface elevation difference trends for *land* (red) and *ice* (blue) samples, respectively, for autumn campaigns 2003–2008. Left: per-tile and -elevation corrections ($c_{tile}$, $c_H$) applied, 1'233 samples; right: also per-glacier correction ($c_{glac}$) applied, 1'105 samples. Trends are computed from individual dh samples using robust linear regression. Campaign median and mean +/- standard error per campaign and class are shown to indicate the variability in dh per campaign. The grey spread corresponds to the measured range of cumulative surface mass balances of 8 glaciers in the area, re-converted to ice volume changes using a density of 850 kg m$^{-3}$ (Andreassen et al., 2016), and their area-weighed mean. The data provided by NVE are based on in-situ and geodetic measurements.

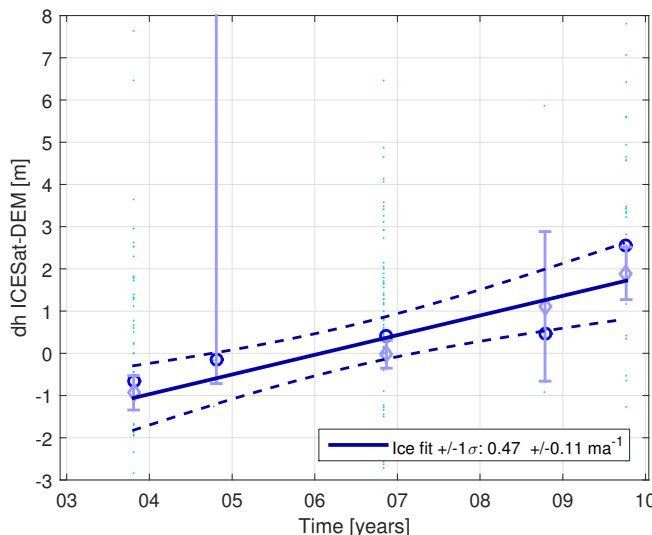

**Figure 7.** The autumn 2003–2009 trend for samples only on those glaciers that are covered by the autumn 2009 campaign (Myklebustbreen and Haugabreen) is strongly positive. The large error bars in 2004 and 2008 result from the very low campaign sample numbers of only 3 and 7 samples, respectively.

