# Peer review of "ICESat laser altimetry over small mountain glaciers"

_The Cryosphere, 2015_

## Referee Comment (RC1) · Anonymous Referee #1 · 8 Mar 2016

The manuscript of the Treichler and Kääb discusses elevation changes of the southern norwegian glaciers, based on differencing of DEM and ICESat data. Their main finding is that, after applying a number of corrections, this method results in credible elevation changes, but the accuracy is limited by uncertainties in the DEM reference data. Although the results and some of the input data (DEMS) are specific to this area, and the negative trend of the southern Norwegian glaciers has been reported elsewhere, I believe this manuscript deserves to be published, after a number of minor changes, as it provides a good road map for future studies using similar methods and that will run into the same limitations of the DEM reference data.

- The authors choose to estimate elevation changes using the DEM differencing approach. However, another, popular method to estimate dh/dt is the plane-fitting approach (Howat 2008 and the Moholdt papers). Please provide a motivation why you

prefer the DEM-ICESat differencing approach for this study.

- The third research question, 'What is the minimum region size w.r.t. glacier density for ICESat GLAS data to ensure statistically significant results' isn't really answered. You do show that you can retrieve an elevation change signal for Myklebustbreen, but don't give a hard lower limit for the region size.

- when assessing elevation bias and spatial shifts between ICESat and the reference DEMs (page 6, section 3.2), did you take into account vertical uplift due to GIA? Uplift rates are in the order of 5 mm/yr in Norway, so over a period of ~40 years (1978-2016), this would result in about 20 cm difference.

- when determining the c_glac correction (page 6, line 31- …) did you check that the temporal coverage by ICESat is sufficient? If a glacier has only been sampled by one ICESat overpass, it's still possible to compute a c_glac correction, but the resulting dh will be ~zero after applying the correction.

- page 13: the c_glac seems only to work if the DEM subset covering the glacier is based on data from one acquisition date (as you also point out on page 14 when discussing the Swiss Alps DEMs). It's worthwhile to point this out here.

- page 10, lines 28-33: do to the increasing cumulative uncertainty in the in-situ mass balance measurements, it's hard to verify this claim. It would be helpful to include a 'mean' in-situ mass balance curve (after applying some weighting to ensure this 'mean' is representative).

- page 11, lines 1-4 + figure 4: the upward jump in the 2009 campaign data is probably an artifact of poor sampling, but what does the in-situ data tell about this year?

- page 13, lines 7: The trends for winter ice samples are indeed more negative, but the uncertainty is much larger, due to the interannual variability in accumulation, and differences with the autumn trends are non-significant. This should be pointed out.

- page 13. limes 14-19: Whether or not the derived trends for such small glaciers are

[Figure]

to be trusted depends to a large extent on the spatial sampling of the glacier. Samples across the entire elevation range are required, with a sampling density resembling the hypsometry distribution of the glacier. Without a further analysis it's impossible to tell what the 0.47 +/- 0.11 m/yr trend represents. Please discuss this in the manuscript.

- figure 7: the uncertainties for '05 are huge. Did something go wrong during plotting, or are these real (if so, it deserves to be discussed in the manuscript).

Technical/minor comments:

Page 1, lines 15-22: I would move this part of the abstract to line 13 (after, "rather than ICESat uncertainty"). Right now you're first discussing the DEM biases, then the ICESat elevation changes and then move back to the DEM biases.

Page 2 line 10: Slobbe 2008 discusses the Greenland Ice Sheet, bot ice caps, so technically, it doesn't belong in this list.

Page 2 lines 20-34: I suggest to use bullet points here to present the list of research questions.

Page 4, line 14: include references for previous ICESat studies

page 5, line 30: start a new paragraph after "... removing footprints on clouds (false positive dh).".

Figure 2: the dotted line is really hard to distinguish (both on screen and print copy)

page 8, line 2: what's the number of ice samples in the autumn 2003 campaign?

Page 9 line: on average

page 16, line 4: change 'volume loss' to 'elevation change' (or convert the -0.34/-0.27 m/yr height changes to volume changes)

---

## Referee Comment (RC2) · R.C. Lindenbergh (Referee) · 10 May 2016

**R.C. Lindenbergh (Referee)**

r.c.lindenbergh@tudelft.nl

Received and published: 10 May 2016

The authors use ICESat satellite laser altimetry elevations as available from 2003 to 2008/2009 to estimate glacial elevation change of small mountain glaciers in Norway. The authors consider several angles to this problem. First, part of the paper could be read as a report on how to extract such glacial elevation changes from the relatively sparse available ICESat elevations over the Norwegian glaciers with the help of locally and globally available auxiliary Digital Elevation Model (DEM) data. An important second angle the authors consider is the influence of the required reference DEM and its possible misalignment on the quality of the results. A third angle, as also the title suggests, is an assessment of using ICESat elevations in general to estimate elevation changes of small mountain glaciers, as can be found all over the world. For this angle it is crucial to assess to what extend local and sparse glacial elevation changes are

representative for a glacial area as a whole.

The paper has valuable contents that are interesting for a larger audience. Small mountain glaciers are present at many different locations on Earth. Monitoring their elevation change by satellite laser altimetry data from ICESat-1 and maybe later ICESat-2 using an additional reference DEM is useful, if detected changes are indeed representative. My problem with the paper in its current state is its focus. If the paper is meant to guide how to extract glacial elevation changes for arbitrary mountain glaciers, at least an analysis on how ICESat is sampling glaciers as a function of latitude is missing: Norwegian latitudes are still relatively favorable, compared to e.g. many South American latitudes. The influence of DEM misalignment is clearly assessed in the manuscript, but how to identify and correct for such misalignment has already been discussed in existing articles. Therefore I suggest to focus the paper on the particular case the authors consider: detecting glacial elevation changes using ICESat and a reference DEM over small Norwegian mountain glaciers. Still, the discussion chapter could be used to generalize to other small mountain glaciers. In addition, the authors should address the following aspects: they don't distinguish between ICESat footprints sampling snow and ice. This should be discussed, and, the effect of this choice on the results should be assessed. The state of the glaciers during ICESat passes could be assessed using additional spectral data or by considering the raw ICESat full waveform signals. Similarly, there might be an effect of terrain roughness and slope on the results, which is not discussed. In addition, the authors confuse glacial elevation change with mass balance change, which are two different things. The authors should discuss why glacial elevation change can directly be linked to mass balance change, notably when one doesn't distinguish between ICESat footprints over snow and ice. Some more detailed remarks are given below.

**Detailed Remarks:**

1. As above: I would focus the paper on Norwegian glaciers, which should be reflected by the title.

2. p2r26: A more general question that is still open: "Is ICESat track density (in combination with average cloud cover) high enough for sparse glaciers at arbitrary latitudes?"

3. p3r20: "two to three month-long observation periods", you mean "two to three observation periods each year of about one month each"

4. p3r21: "42 km" this may hold for Norway, but is in general latitude dependent.

5. p4r14: "ICESat tracks of more than one year": funny English, please reformulate.

6. Section 2.3: what are the difference between: "vertical accuracy", "mean error" and "standard error", please define these notions.

7. p5r4: -> "The 2009 autumn campaign is excluded" (skip 'usually' to avoid confusion)

8. p5r9: what is the influence of the 40m threshold for "ice border"? Apparently (Section 3.3) this threshold has a strong influence on the amount of ICESat elevations that are considered to fully cover glaciers (given the quotes of 2.5% on glacier points, and 0.9% of border points)

9. p5: "snow heights": (K"a"ab, 2012;2015) discusses Central-Asian glaciers. Why can conclusions on snow variations there be simply ported to a Norwegian setting? And would there be no big differences for valley glaciers compared to icefields, as this figure of Jostedalsbreen suggests: https://en.wikipedia.org/wiki/Jostedal\_Glacier#/media/File:P1000290Jostedalsbreen.JPG Why is it not actively assessed if glaciers are covered by snow at the time of the ICESat passes? That could also assist in the issues on winter snow fall and December campaigns raised in Section 4.4.

10. p5, IWD, what parameter? I.e. what power?

11. p5: how did the outliers look like that were removed by the robust fitting? How did the spatial pattern of cloud affected ICESat elevations look like?

12. p6r1: can you quantify "larger number of outliers"?

13. p6: did you experience any issues in the LIDAR data due to not fully adjusted flight strips? (Remaining errors after strip-adjustment)

14. p6: what are possible reasons for the shifts in the Kartverkets DEMs?

15. Section 4.1: how do you know the dh are t-distributed?

16. Saturation may occur along track when ICESat hits bare ice after rock (as it takes the gain  $\sim$ 5 shots to reset after hitting the more reflective ice). Did you consider the spatial distribution of the saturated waveforms? (compare Molijn RA, Lindenbergh RC, Gunter BC. ICESat laser full waveform analysis for the classification of land cover types over the cryosphere. International journal of remote sensing. 2011 Dec 10;32(23):8799-822)

17. You state: "However, these differences cancel out (Fig4)". Could you help the reader seeing that in Figure 4.?

18. From the material just in this paper it is difficult to understand what you mean by p10r25-27: "This stresses...weight". Could you explain this a bit more extensively?

19. p12r6: you say "terrain characteristics" are essential, but, as argued by me before you only consider these only in a very global way.

20. Section 5.3: do you believe that indeed the age of the DEM is crucial, or rather the way it was constructed (photogrammetry, radar, LIDAR)?

21. p15r13: saturation correction (and other flags). I would say this is an interesting topic for more study, to check how the rapid transitions between land-cover on small mountain glaciers influence the ICESat raw signal (and its corrections)

22. p15: quality of the reference DEM vs ICESat: should it not be only the quality, but also its spatial resolution compared to the ICESat footprints compared to the local relief variations?

СЗ

---

## Author Comment (AC2) · 6 Jul 2016

**Response to referee comments**

**ICESat laser altimetry over small mountain glaciers**

D. Treichler and A. Kääb

**We would like to thank the two reviewers for their constructive feedback and valuable input that certainly helped to improve the article. Detail responses are provided below, together with a mark-up manuscript version where the changes made in response to the referees' comments are highlighted.**

**Anonymous Referee #1**

*The manuscript of the Treichler and Kääb discusses elevation changes of the southern Norwegian glaciers, based on differencing of DEM and ICESat data. Their main finding is that, after applying a number of corrections, this method results in credible elevation changes, but the accuracy is limited by uncertainties in the DEM reference data. Although the results and some of the input data (DEMS) are specific to this area, and the negative trend of the southern Norwegian glaciers has been reported elsewhere, I believe this manuscript deserves to be published, after a number of minor changes, as it provides a good road map for future studies using similar methods and that will run into the same limitations of the DEM reference data.*

*- The authors choose to estimate elevation changes using the DEM differencing approach. However, another, popular method to estimate dh/dt is the plane-fitting approach (Howat 2008 and the Moholdt papers). Please provide a motivation why you prefer the DEM-ICESat differencing approach for this study.*
**The plane-fitting approach requires constant surface slopes between neighbourhoods of footprints as found over large ice bodies and thus fails in rough mountain terrain. Using a reference DEM instead accounts for the more complex topography of mountain landscapes. The motivation for our choice including references to studies using the plane-fitting approach has been added in section 3.**

*- The third research question, 'What is the minimum region size w.r.t. glacier density for ICESat GLAS data to ensure statistically significant results' isn't really answered. You do show that you can retrieve an elevation change signal for Myklebustbreen, but don't give a hard lower limit for the region size.*
**The reviewer is correct that the manuscript only indirectly deals with that question. We found that there is no hard lower limit for a region size but the area/statistical sample size required depends on the combination of glacier/ICESat track density, homogeneity of the glacier signal, and sample representativeness, i.e. factors specific to each area – in that regard, the question may not be a good one to ask in the first place as there is only a qualitative answer to it. Since it is nevertheless**

asked frequently by users of our method we decided not to remove it. We added a new paragraph in section 5.1 that emphasises the factors that need to be considered when grouping ICESat glacier footprints spatially. With this, we hope to provide a more satisfactory and direct – although still relative – answer to the minimum region size.

*- when assessing elevation bias and spatial shifts between ICESat and the reference DEMs (page 6, section 3.2), did you take into account vertical uplift due to GIA? Uplift rates are in the order of 5 mm/yr in Norway, so over a period of _40 years (1978-2016), this would result in about 20 cm difference.*

We did not consider vertical uplift due to GIA – the up to 20cm uplift the reviewer correctly calculated are much smaller than median tile/date offsets (up to +/-1m per tile, and +/- 5m per date) or elevation-dependent error cH (dm per 100m elevation). Also, we could not find a relationship between the DEM age (proxy) and elevation bias, leaving us with no evidence of a glacial rebound effect on the ICESat-Kartverket DEM elevation differences on stable terrain. On glaciers, the c_glac correction removes all offsets regardless of their origin. A note on the magnitude of GIA uplift compared to ICESat-reference DEM elevation differences has been added in section 3.2.

*- when determining the c_glac correction (page 6, line 31- : : :) did you check that the temporal coverage by ICESat is sufficient? If a glacier has only been sampled by one ICESat overpass, it's still possible to compute a c_glac correction, but the resulting dh will be _zero after applying the correction.*

The reviewer is correct that for glaciers that only experience one overpass (possible due to spatial variability of ICESat ground tracks) average differences will be zero. A priori, it can be assumed that mostly small glaciers (i.e. very few samples) would be affected by the problem of a single overpass only. The expected effect of the zero differences is to flatten out the trend. In our study we saw the contrary – surface elevation trends became steeper after application of c_glac. One can argue that the bias introduced by these samples is likely smaller than the considerably larger offsets we found from DEM age and vertical offsets, and that it should be captured by the trend confidence interval. However, the introduced bias from these single overpass samples is systematic, unlike other (random) error sources – especially if they occur in the beginning or end of the acquisition period. We concluded that this bias should be assessed and quantified, following the reviewers concerns.

A thorough analysis on the samples in question revealed the following:
- **128 samples on 36 glaciers are from single ICESat overpasses (only autumn campaigns), corresponding to ca. 10% of ice samples and glaciers hit.**
- **90% of the glaciers sampled only during one campaign (27 glaciers) were sampled in autumn 2003. This can be explained by the transition from an 8 day repeat orbit that was flown for the first 10 days of the autumn 2003 campaign to the 91 day repeat orbit used from October 4th, 2003 through the entire ICESat mission (Schutz et al., 2005). The two orbit repeat cycles have different ground tracks, resulting in 113 of 427 samples falling on glaciers sampled only by a single overpass for this particular campaign. A closer examination of the glaciers in question showed that a large fraction of the samples in question are from an overpass over northern**

Folgefonna where our DEM age proxy indicates reference elevations from the 1980ies. The
(uncorrected) dh of these samples are considerably more negative. The option of not applying
c_glac to one overpass samples (but all other samples) is therefore not a good solution in our
case.
–  The other 15 single overpass samples are well distributed between the remaining campaigns as
well as in space and with regard to reference DEM age.
–  Exclusion of the 128 one-overpass glacier samples results in a slight shift towards larger dh of
the median (but not the mean) of the 2003 autumn campaign and, consequently, slightly
increases the trend slope (Figure 1). The effect this has on the trend slope lies within the trend
standard error and is considerably smaller than the effect of applying c_glac in the first place.
–  The trend slope difference corresponds to a steepening of 0.05 ma$^{-1}$ for all subsets – except for
where sample numbers are considerably smaller and especially where glacier size plays a role:
glaciers <5km2, pre-2000 DEM source date and East of water divide. For the latter two the
glacier size is an indirect cause (more small glaciers further east and outside areas where DEM
updates were prioritised). There, the increase in trend slopes is > 0.1 ma$^{-1}$.This confirms how
sensitive trends are to bias in dh – especially when sample numbers are small.

**In the revised manuscript we consider these new insights throughout the text, specifically in**
**sections 4.1 and 4.4 in the results, in the discussion, as well as in Table 1.**

[Figure]

**Figure 1:** Glacier surface elevation trends with (left panel) and without (right panel) *ice* samples on glaciers sampled by a
single overpass only.

*- page 13: the c_glac seems only to work if the DEM subset covering the glacier is based on data from*
*one acquisition date (as you also point out on page 14 when discussing the Swiss Alps DEMs). It's*
*worthwhile to point this out here.* **DONE on page 10 (assuming a typo in the page number above)**

*- page 10, lines 28-33: do to the increasing cumulative uncertainty in the in-situ mass balance*
*measurements, it's hard to verify this claim. It would be helpful to include a 'mean' in-situ mass*
*balance curve (after applying some weighting to ensure this 'mean' is representative).*
**The range of cumulative glacier surface balances measured by NVE has been updated in the revised**
**manuscript, using the harmonised/calibrated data that is now available from NVE (NVE, 2016;**

**Andreassen et al., 2016). A mean cumulative mass balance curve for these glaciers (weighed by**

**respective glacier areas) has been added to the figures. The new harmonised data fit the ICESat**

**trend even better than the non-harmonised data shown in the discussion manuscript (Figure 2).**

**Note that in the corresponding figure in the revised manuscript, only 8 (instead of 10) in-situ**

**glacier series are included as mass balance measurements on Folgefonna outlet glaciers started**

**only in 2006, and homogenised data for these very short and recent time series are not yet**

**distributed by NVE.**

[Figure]

**Figure 2:** ICESat glacier elevation trend compared to surface mass balance (smb) of 8 glaciers in southern Norway measured by NVE (based on in-situ data and geodetic methods). Shown are the range of cumulative smb re-converted to ice volume using a density of 850 kg m$^{-3}$ (Andreassen et al., 2016), and their weighed mean (solid line, weighed by glacier area) of the newly available harmonised data, as well as the area-weighed mean of the volumetric balance of 10 glaciers before harmonisation (dotted line, corresponding to the data shown in the grey spread in the corresponding figure in the discussion manuscript).

*- page 11, lines 1-4 + figure 4: the upward jump in the 2009 campaign data is probably*

*an artifact of poor sampling, but what does the in-situ data tell about this year?*

**From in-situ data we expect a slightly negative balance in 2009. A comment stressing this**

**discrepancy has been added to section 4.4.**

*- page 13, lines 7: The trends for winter ice samples are indeed more negative, but the uncertainty is*

*much larger, due to the interannual variability in accumulation, and differences with the autumn*

*trends are non-significant. This should be pointed out.*   **DONE**

*- page 13. limes 14-19: Whether or not the derived trends for such small glaciers are to be trusted*

*depends to a large extent on the spatial sampling of the glacier. Samples across the entire elevation*

*range are required, with a sampling density resembling the hypsometry distribution of the glacier.*

*Without a further analysis it's impossible to tell what the 0.47 +/- 0.11 m/yr trend represents. Please*

*discuss this in the manuscript.*

**The reviewer is absolutely correct that also such a local trend is only valid if the ICESat samples are**

**representative for the glaciers in question. This is the case here (see Figure 3), and the**

**representativeness of this sub-sample is now mentioned in section 4.4 and discussed in a more**
**general way in the new paragraph in section 5.1 (see also reply to the reviewer's second comment).**
**However, we would like to emphasise that in our study, the main role of mentioning the**
**Myklebustbreen/ Haugabreen surface elevation trend lies in explaining the 2009 jump and in**
**stressing the need for per-campaign representativeness also in terms of good and consistent**
**spatial distribution. In that sense it is a side product of our study. While we have no reason to**
**assume that the trend is wrong we strongly advise that the surprisingly positive glacier surface**
**elevation change trend on Myklebustbreen/Hansebreen is critically reviewed and, as far as**
**possible, verified with other data in case it should be used in further studies.**

[Figure]

**Figure 3:** Representativeness of Myklebustbreen and Haugabreen (normalised frequency in %; elevation in m, slope in
degrees, aspect in degrees from North). The dotted line corresponds to all DEM cells of these glaciers. Note that campaigns
2005 and 2007 contain none and campaigns 2004 and 2008 only 3 and 7 samples, respectively, reflected in worse fit of
these curves compared to the ICESat full sample/entire glacier area from the reference DEM for these glaciers.

*- figure 7: the uncertainties for '05 are huge. Did something go wrong during plotting, or are these*
*real (if so, it deserves to be discussed in the manuscript).*
**The reviewer's comment concerns the trend for Myklebustbreen/Haugabreen. The autumn 2004**
**campaign (closer to the 2005 axis tick) consists of only 3 samples on these glaciers of which one**
**obviously is an outlier with very large dh, resulting in a huge standard error of the campaign mean**
**(error bar). A comment on these low campaign sample numbers has been added to section 4.4 and**
**in the figure caption.**
*Technical/minor comments:*
*Page 1, lines 15-22: I would move this part of the abstract to line 13 (after, "rather than ICESat*
*uncertainty"). Right now you're first discussing the DEM biases, then the ICESat elevation changes*
*and then move back to the DEM biases.*   **DONE**

*Page 2 line 10: Slobbe 2008 discusses the Greenland Ice Sheet, bot ice caps, so technically, it doesn't belong in this list.*

**The reference has been removed from the list.**

*Page 2 lines 20-34: I suggest to use bullet points here to present the list of research questions.* **DONE**

*Page 4, line 14: include references for previous ICESat studies* **DONE**

*page 5, line 30: start a new paragraph after "... removing footprints on clouds (false positive dh).".*
**DONE**

*Figure 2: the dotted line is really hard to distinguish (both on screen and print copy)*
**The figure has been updated to better distinguish the land spread. Note that the land spread may still be difficult to see as it is very narrow, and that both ice/land spread have slightly changed shapes in the updated manuscript since the autumn 2009 campaign (excluded from analyses) is now also excluded from the spread. This was, erroneously, not the case before.**

*page 8, line 2: what's the number of ice samples in the autumn 2003 campaign?*
**427 ice samples, this information has been added**

*Page 9 line: on average* **DONE**

*page 16, line 4: change 'volume loss' to 'elevation change' (or convert the -0.34/-0.27 m/yr height changes to volume changes)* **DONE**

**R.C. Lindenbergh**

*The authors use ICESat satellite laser altimetry elevations as available from 2003 to 2008/2009 to estimate glacial elevation change of small mountain glaciers in Norway. The authors consider several angles to this problem. First, part of the paper could be read as a report on how to extract such glacial elevation changes from the relatively sparse available ICESat elevations over the Norwegian glaciers with the help of locally and globally available auxiliary Digital Elevation Model (DEM) data. An important second angle the authors consider is the influence of the required reference DEM and its possible misalignment on the quality of the results. A third angle, as also the title suggests, is an assessment of using ICESat elevations in general to estimate elevation changes of small mountain glaciers, as can be found all over the world. For this angle it is crucial to assess to what extend local and sparse glacial elevation changes are representative for a glacial area as a whole.*
**We agree with the referee that the paper deals with several aspects of ICESat over mountain glaciers and readers might read it with a different focus in mind. Our motivation for this study was to thoroughly assess ICESat-derived glacier surface elevation changes – a method that has already**

**been applied globally in many places and thus begins to be relatively established, but has so far**
**not been thoroughly validated. In that sense, we see our contribution as a road map to improve**
**ICESat applications on small mountain glaciers in general – not only in Norway. This was reflected**
**in the title, as the reviewer mentioned, too.**
*The paper has valuable contents that are interesting for a larger audience. Small mountain glaciers*
*are present at many different locations on Earth. Monitoring their elevation change by satellite laser*
*altimetry data from ICESat-1 and maybe later ICESat-2 using an additional reference DEM is useful,*
*if detected changes are indeed representative.*
*My problem with the paper in its current state is its focus. If the paper is meant to guide how to extract*
*glacial elevation changes for arbitrary mountain glaciers, at least an analysis on how ICESat is*
*sampling glaciers as a function of latitude is missing: Norwegian latitudes are still relatively*
*favorable, compared to e.g. many South American latitudes. The influence of DEM misalignment is*
*clearly assessed in the manuscript, but how to identify and correct for such misalignment has already*
*been discussed in existing articles. Therefore I suggest to focus the paper on the particular case the*
*authors consider: detecting glacial elevation changes using ICESat and a reference DEM over small*
*Norwegian mountain glaciers. Still, the discussion chapter could be used to generalize to other small*
*mountain glaciers.*
**The selection of Norway as a test site was made due to the reference data available here, a key**
**condition for a solid method assessment. We are certain – and this is also stressed in our**
**manuscript – that the issues discussed (representativeness, DEM quality) are transferrable to other**
**locations. The study builds on (and is also motivated by) extensive tests done on ICESat**
**applications in High-Mountain Asia (Kääb et al, 2012; supplement). While based on a method**
**assessment in Norway, our findings are not only specific to Norway. We very intentionally analyse**
**our results from Norway with a broader, more global horizon in mind in the discussion section.**
**We agree with the reviewer that ICESat sampling in relation to latitude would certainly show that**
**higher latitudes are more favourable. At the same time, latitude does only to some degree affect**
**whether or not ICESat applications on glaciers are possible or not in a given region. Other factors –**
**glacier density, size, position in relation to ICESat tracks, and homogeneity of the glacier signal –**
**are equally important, if not more so. The key requirement is the representativeness of the**
**samples, and this has to be assessed locally. We believe that a visualisation of ICESat glacier**
**samples vs. latitude would give this particular aspect of ICESat applicability too much weight in**
**comparison with other factors to consider. We prefer to keep the focus of the paper on the**
**method, not on identifying the optimal places for ICESat glacier studies on the entire globe.**
**In the revised manuscript, we try to stress the dependence of track density on latitude more to**
**make the reader aware of this fact. That this is not the only factor to consider when applying our**
**method is now better stressed in a new paragraph on representativeness and minimum region size**
**in section 5.1 in the revised manuscript.**
**We certainly don't claim that DEM misalignment is a new discovery of ours, but we emphasise that**
**the commonly done global co-registration of entire DEM tiles with ICESat samples may not remove**
**all bias. The fact that shifts of spatially unknown DEM sub-units can have a strongly biasing effect**
**on ICESat analyses, as well as our proposed localised correction (c_glac), are new. We feel that the**

issues we discuss have so far not received enough attention by users of the method and hope to
make users of ICESat in other glacierised areas aware of them with our contribution.
We are afraid that the article would not reach all of the potentially interested scientific audience if
the focus lay on Norwegian glaciers. These glaciers are well studied and our ICESat surface
elevation trends only confirm the results from in-situ and geodetic mass balance studies. However,
we agree with the structure the reviewer proposes: results from Norway, but discussed in a
general, global context. We tried to make this now even clearer in the manuscript. We did not
change the focus to Norwegian glaciers only, as the reviewer suggested, but kept it global/general.
We hope the reviewer agrees.
*In addition, the authors should address the following aspects: they don't distinguish between ICESat*
*footprints sampling snow and ice. This should be discussed, and, the effect of this choice on the results*
*should be assessed.*
The reviewer is correct in that we don't distinguish between ICESat footprints on parts of the
glacier that are snow-covered/bare. This is intentional: In order to capture a signal that may be
related to geodetic mass balance we need to sample the entire glacier to consider both surface
elevation changes from ice melt and glacier dynamics. Footprints on only snow-covered or bare ice
parts of the glaciers would likely only lie on the accumulation or ablation parts of the glaciers,
respectively, and do thus not fulfil the condition of mass continuity. It would therefore be
physically incorrect to draw conclusions on the glaciers' mass balance from a trend based on such a
subset of samples. This is an important and inherent condition of any volumetric-geodetic glacier
method, including ICESat studies. It is also part of the reason why we stress the need for
representativeness so much. An explanation on this has been added to section 3.3 where sample
subsets are introduced.
The referee might also point to potentially different densities over ice and snow to convert
elevation changes to mass changes. This issue is a tricky one as one cannot strictly know if an
elevation change is due to a change in the ice column or in snow or firn thickness, new firn, or if
the density profile changed over time (firn compaction, superimposed ice, etc.). These issues are
discussed in Huss (2013) which we rely on for our density assumption. Further, ICESat dh density
scenarios have also been evaluated in the cited Kääb et al. (2012). See also below response on
density, and volume vs. mass.
*The state of the glaciers during ICESat passes could be assessed using additional spectral data or by*
*considering the raw ICESat full waveform signals.*
The inclusion of additional remote sensing data to characterise ICESat footprints and the surface
they fall on is a good idea that we very much support. We agree that for a follow-up study with a
focus on seasonal changes or mass turnover such a distinction might be valuable. While we argue
above why we think this is not appropriate in the way the reviewer proposes for the current study,
it is to some degree also not possible, due to the following reasons:
-    Within optical remote sensing data, Landsat would be the most promising candidate to provide
continuous data for land cover classification in Norway in 2003-2009. However, Landsat's
repeat cycle of 16 days, combined with Norway's rather cloudy weather, makes it
unfortunately impossible to ensure cloud-free coverage for operational classification of ICESat

**footprints on sufficient temporal time scale to reliably detect snow events and snow cover.**
**MODIS fails as an alternative due to its coarse spatial resolution. A combination of both**
**datasets complemented with modelled data, such as the snow cover maps distributed by the**
**Norwegian Water and Energy Directorate NVE (available on senorge.no), could possibly work,**
**although large uncertainty from interpolation and other modelling aspects would have to be**
**expected.**
- **Our impression from previous studies are that waveform classification methods may be used**
**to classify land cover types in general but seem to be struggling to reliably distinguish different**
**glacier surface types with sufficient accuracy (Molijn et al., 2011; Shi et al., 2013). We think**
**these methods would need to be further refined to be used in a more operational way, i.e.**
**without adding large uncertainties. Waveform analyses would be most interesting for**
**applications where distinction of within-footprint snow cover or fine-scale surface topography**
**are crucial. A possible approach to increase classification accuracy could be waveform fitting to**
**within-footprint topography from a reference DEM. Snow cover (smoothing effect) might be**
**detected where waveforms and reference DEM surface or other indicators (likelihood of**
**smooth/rough surface from timing, elevation and slope of footprint) don't match.**
**Footprint classification with the above methods would add uncertainty to our analyses that would**
**be hard to quantify and, consequently, make validation of the method more difficult – which is the**
**prime focus of the work. We prefer therefore not to do so in this study but see this as an**
**interesting idea for further work.**
**As a side note, we did already a study on modelling waveforms from high-res elevation models and**
**ground reflectivity data, and we analysed waveforms over rough mountain topography. Both**
**unpublished studies didn't lead to very conclusive results that would clearly benefit the study**
**under discussion here. Most likely, the rough and variable topography over mountains and**
**mountain glaciers make it difficult to retrieve simple rules based on waveforms that could be**
**applied to regional (i.e. not local) mountain studies.**
*Similarly, there might be an effect of terrain roughness and slope on the results, which is not discussed.*
**We agree with the reviewer that terrain roughness and slope will alter the ICESat return waveform**
**and thus might affect extracted elevations. Already (Kääb et al., 2012; article supplement) did a**
**thorough analysis on the potential effect of slope on ICESat elevations in the Himalayas. There, the**
**most relevant finding was a positive relation between saturation of footprints and slope in**
**particular for off-glacier terrain. It remains unclear which way round the causality works – sloping**
**terrain causes saturated footprints, or the saturation classification algorithm misclassifies these**
**waveforms as saturated due to their shape.**
**Also in Norway, we did extensive tests to discover potential systematic bias on dh from slope or**
**within-footprint surface topography, using the standard deviations of slope, aspect and elevation**
**of 10m DEM grid cells within an assumed 70m circular ICESat footprint as a proxy. We found no**
**significant relationship between dh and slope or within-footprint topographic roughness, and no**
**indication for a systematic bias from footprint slope or roughness, also not in combination with**
**waveform saturation. Note that the Norwegian mountains present a different landscape than the**
**Himalayas, and we experienced both much lower numbers of saturated samples and flatter slopes**
**in this study than in the Himalayas, in particular for land samples.**

**Still, the reviewer is correct in that the slope of the footprints has an influence on derived trends.**
**Bootstrapping methods on sample subsets using different slope thresholds show that ICESat glacier**
**surface elevation trends in southern Norway are more negative for samples on larger slopes.**
**However, it would be wrong to draw the conclusion that steeper glaciers experience stronger melt**
**than flatter ones in our study area, as the phenomenon is explained by topography and glacier**
**physics: Many of southern Norway's glaciers are small ice caps and have thus relatively flat**
**accumulation areas on top of the rounded mountains, smoothened by the Scandinavian ice shield**
**(an example is Jostedalsbreen ice cap the reviewer refers to below). Glacier tongues, on the other**
**hand, extend into the steep fjords and valleys. A sample subset based on slope resembles**
**therefore a sample subset of elevations. Trends derived from samples at high/low elevations (i.e.**
**accumulation/ablation parts only) reveal the same effect, but even stronger: At high elevations,**
**surface elevation change trends are considerably flatter than at elevations of the glacier tongues.**
**This is a direct effect of glacier flow physics. Considering the fact that we see a trend at all we can**
**be sure that the glaciers in southern Norway were not in balance between 2003 and 2008.**
**Negative imbalance may result from increased melt – more pronounced at lower, warmer**
**elevations – or decrease of precipitation. In both cases, glacier flow will eventually transport**
**changes over the entire glacier due to ice flow, i.e. increased melt at the tongue will also result in**
**lowering of the accumulation areas. The signal may though be delayed, and the interpretation of**
**such non-trivial differences in surface elevation changes of different glacier parts is beyond the**
**scope of this study.**
**Our goal with this study is to validate and improve the method of deriving volumetric glacier**
**surface elevation changes from ICESat elevations. Consequently, we do not discuss aspects that**
**were found to have a minor or no influence on derived trends in detail but rather focus on the**
**main influencing/biasing factors.**

**In the revised manuscript, we added more explanations in section 3.3 and 4.4 to ensure readers**
**are aware of glacier mass continuity where sample representativeness and sample subsets are**
**discussed.**

*In addition, the authors confuse glacial elevation change with mass balance change, which are two*
*different things. The authors should discuss why glacial elevation change can directly be linked to*
*mass balance change, notably when one doesn't distinguish between ICESat footprints over snow and*
*ice. Some more detailed remarks are given below.*
**See also above response on density.**
**Our study confirms that ICESat-derived glacier surface elevation changes indeed accurately reflect**
**volumetric glacier balance – which may thereafter be converted into glacier mass balance with the**
**use of ice/snow/firn densities. Since the focus of this particular study lies on the method (which is**
**inherent to ICESat and contributing factors) and not on conversion between volumetric and mass**
**changes (which applies also to e.g. DEM differencing, and which does not depend on ICESat**
**parameters but local conditions mainly), we prefer not to discuss the problem of volume/mass**
**conversion here. In contrast to e.g. Kääb et al. (2012) or Gardner et al. (2013) whose focus lay on**
**the derived trends, not on the method itself, we do not attempt to convert ICESat surface**

**elevation trends to mass changes and draw conclusions thereof. We only relate ICESat trends to**
**mass changes to compare to the NVE mass balances and use their choice of density for this**
**purpose, i.e. in a way just back-convert NVE's results. We checked again and tried our best to be**
**very clear and consistent in our related vocabulary throughout the manuscript.**

**The results of Kääb et al. (2012) differ by only 5% between two different density scenarios for**
**conversion of ICESat elevation trends to mass change. One of the two applied scenarios**
**distinguishes between firn and ice areas (600 and 900 kg m$^{-3}$, respectively) while the other assumes**
**an average density of 900 kg m$^{-3}$. (Gardner et al., 2013) also assume a density of 900 kg m$^{-3}$. To**
**compare the in-situ data with ICESat-derived elevation changes, we back-converted water**
**equivalent to ice assuming a density of 850 kg m$^{-3}$ based on the findings of Huss (2013) – which also**
**NVE used for their geodetic data – and which can be seen as a new standard value for glacier**
**volume/mass conversion for volumetric-geodetic mass balance studies. While the conditions for**
**this number are not perfectly in place – given the imbalance and year-to-year variation of southern**
**Norway's glaciers that indicates instable mass gradients – it should serve sufficiently well as a best**
**guess for the validation purpose the in-situ data has in our study. A paragraph that highlights the**
**difference between ice surface elevation and mass changes, and that justifies this choice more**
**explicitly, has been added to section 3.**

*Detailed Remarks:*
*1. As above: I would focus the paper on Norwegian glaciers, which should be reflected*
*by the title.*
**As explained above, we prefer to keep the title short and attractive for readers that should find**
**relevant information for their work. And we believe the information given is by far not only**
**relevant for Norwegian glaciers but rather for ICESat over mountain glaciers in general.**

*2. p2r26: A more general question that is still open: "Is ICESat track density (in combination*
*with average cloud cover) high enough for sparse glaciers at arbitrary latitudes?"*
**We agree with the reviewer that the reader might like to get an answer also on more general**
**questions, such as the main factors that govern ICESat applicability in glacierised regions. While we**
**discuss many of these factors already now, the corresponding question is currently missing in the**
**manuscript. Since our findings show that latitude plays an indirect role in the way that it affects**
**sample numbers and, subsequently, representativeness, we formulated a new question that is**
**even more general:** *What prerequisites and conditions need to be fulfilled to make ICESat-derived*
*elevation changes over a certain area a valid method to assess glacier volume changes?*
**This additional research question is discussed together with the minimum region size in a new**
**paragraph in section 5.1**

*3. p3r20: "two to three month-long observation periods", you mean "two to three observation*
*periods each year of about one month each"* **CHANGED**

*4. p3r21: "42 km" this may hold for Norway, but is in general latitude dependent.*
**We added a note on the relation between cross-track spacing and latitude.**

*5. p4r14: "ICESat tracks of more than one year": funny English, please reformulate.*  **DONE**

*6. Section 2.3: what are the difference between: "vertical accuracy", "mean error" and*
*"standard error", please define these notions..*  **DONE**

*7. p5r4: -> "The 2009 autumn campaign is excluded" (skip 'usually' to avoid confusion)*  **DONE**

*8. p5r9: what is the influence of the 40m threshold for "ice border"? Apparently (Section 3.3) this*
*threshold has a strong influence on the amount of ICESat elevations that are considered to fully cover*
*glaciers (given the quotes of 2.5% on glacier points, and 0.9 % of border points)*
**We are not sure we understand the reviewer's question correctly. The goal of the 40m in- and**
**outside buffer is to avoid footprints falling both on ice and land, i.e. giving a mixed elevation signal.**
**There is little room for discussion for the threshold itself as it corresponds to roughly the diameter**
**of one ICESat footprint (ca. 70m). The reasoning for this has been reformulated in section 3.1 to**
**ensure better clarity. The buffer could potentially be larger to better account for spatial**
**uncertainty of glacier outlines – both from potential changes over time, and from the limits in**
**accuracy due to the fact that glacier outlines usually are derived from satellite data with a given**
**pixel size. This would result in more *ice border* footprints and lower the *ice* sample numbers**
**further. We believe that the 40m buffer is appropriate for the comparably high quality of the**
**glacier outlines in southern Norway. The 0.9% share of *ice border* footprints of all samples reflects**
**the fact that many glaciers in southern Norway are rather small, resulting in a comparably large**
**buffer area around the many small ice bodies.**
**Ice border samples could both exhibit large dh where a glacier has melted/retreated/advanced**
**since the reference DEM was acquired, or very small dh where there never has been ice within that**
**footprint in the first place. The dh distribution of *ice border* samples was found to be wider than**
**for *land* and *ice* dh and is not affected by c_glac correction (Fig. 2 in the manuscript). While one**
**could argue that these large dh indicate changes in ice elevations, inclusion of *ice border* samples**
**in trend calculations affects the trend slope if c_glac is not applied, and increases uncertainty if**
**c_glac is applied (the trend uncertainty is the same as for *ice* samples only, but given the higher**
**sample numbers it should be smaller if the additional samples contributed with dh representing**
**accurate and valid measurements). We therefore advise users to exclude ice border samples also in**
**other areas where the introduced uncertainties may be even larger. This recommendation has**
**been added explicitly in section 5.1.**
**We hope this answers the reviewer's comment.**

*9. p5: "snow heights": (Kääb, 2012;2015) discusses Central-Asian glaciers. Why can conclusions on*
*snow variations there be simply ported to a Norwegian setting?*
*And would there be no big differences for valley glaciers compared to icefields, as this figure of*
*Jostedalsbreen suggests:*
*https://en.wikipedia.org/wiki/Jostedal_Glacier#/media/File:P1000290Jostedalsbreen.JPG*
**The reviewer's comment relates to our argumentation why spring and winter campaigns were**
**excluded from trends. High-Mountain Asia (HMA) includes glaciers in a wide range of different**
**environmental conditions. The argumentation in Kääb et al. (2012; 2015) is a general one that**
**discusses the meaning of the signal and not, as the reviewer might have assumed, applicable to**

**HMA glaciers only. They postulate that yearly varying timing and magnitude of snow fall cannot**
**ensure that the ICESat overpasses measures winter surface balances reliably. Snow densities are**
**expected to vary highly depending on ICESat timing in relation to snow pack evolution. Yearly**
**varying net glacier balance will affect glacier mass turnover which will also be reflected in winter**
**surface elevations. As a consequence, winter elevations measured by ICESat reflect a mix of**
**elevation changes in ice and snow surfaces that is hard to resolve. This is especially the case for**
**time series as short as the five years where ICESat data is available. Cumulative net balances, such**
**as our ICESat glacier elevation trends represent, should therefore be based on yearly net balances**
**from the end of the hydrological year. For method comparability with studies of other authors**
**(Gardner et al., 2013; Ke et al., 2015; Kropáček et al., 2014; Neckel et al., 2014) who include winter**
**data in their ICESat-derived glacier trends, we computed winter trends for our study site, too. Our**
**results are in line with the arguments brought forward already by Kääb et al. (2012; 2015). While**
**winter campaigns (as well as samples from the accumulation/ablation areas, or samples on snow-**
**covered/bare ice only) may contain a signal of changes in mass turnover potentially interesting for**
**a future study, we advise against the use of winter campaigns to derive glacier surface elevation**
**changes where these should be related to/used as glacier mass balance series.**
*Why is it not actively assessed if glaciers are covered by snow at the time of the ICESat passes? That*
*could also assist in the issues on winter snow fall and December campaigns raised in Section 4.4.*
**We refer to our argumentation above for distinguishing snow-covered vs. bare ice samples.**
**Concerning the December campaign: There are to date unfortunately no remote sensing methods**
**that measure snow heights. Optical sensors are only capable to map snow cover. However,**
**modelled snow depths from NVE (available from senorge.no) and observations from**
**meteorological stations (data from the Norwegian Meteorological Institute, available from**
**eklima.met.no) confirm the onset of snowfall in November/December 2008 – which is to be**
**expected at that time of the year in Norway – and suggest the dh measured by ICESat are within**
**plausible range. Since correction of the December 2008 dh with snow depths estimates based on**
**December 2008 *land* samples results in an even flatter *land* trend and no noticeable change in the**
***ice* trend (Table 1, section 4.4), we believe this issue is sufficiently explained and expect no further**
**insight from the use of additional data. We hope the reviewer agrees.**
*10. p5, IWD, what parameter? I.e. what power?*
**Power 1, i.e. linear inverse distance weighing. Information added in section 3.1.**
*11. p5: how did the outliers look like that were removed by the robust fitting? How did*
*the spatial pattern of cloud affected ICESat elevations look like?*
**Robust fitting removed *ice* samples with |dh|>14m (Figure 4). After removal of one-overpass**
**samples (see reply to reviewer 1), 76 of 1105 samples received a weight of 0.3 or lower (coloured)**
**and 26 samples received weight 0. For footprints on slopes > ca. 25 degrees weights decrease with**
**increasing slope (Figure 5) which is in accordance with the larger expected elevation uncertainty of**
**footprints on sloping terrain. There is no clear dependency on footprint elevation, aspect, or area**
**of the glacier sampled (Figure 5), and there is no visible pattern in the spatial distribution of the**
**samples weighed less than 0.3 (Figure 4).**
**A note on the number of samples that received weight 0 has been added to section 3.1.**

**ICESat samples identified as cloud elevations (dh>100m, excluded from trend analyses) have a**

**similar spatial distribution as the samples used in trend analyses (Figure 6).**

[Figure]

**Figure 4:** dh (left) and spatial distribution (right) of *ice* samples in relation to the weights assigned by robust fitting.

[Figure]

**Figure 5:** Weights assigned to *ice* samples by robust fitting as compared to glacier-governing parameters.

[Figure]

**Figure 6:** Spatial distribution of cloud samples (left) that were excluded from the study compared to samples included in the
study (right, map corresponding to fig. 1 in the manuscript).

*12. p6r1: can you quantify "larger number of outliers"?*
**A quantification of the number of "outliers" for any distribution requires a threshold dh or,**
**alternatively, a p-value that corresponds to a fraction of samples considered valid. Figure 7 shows**
**our *ice* dh distribution (c_glac applied) as well as fitted t- and normal distributions. In this direct**
**comparison, it becomes obvious that the dh and t distributions have longer tails than the normal**
**distribution with its bell shape. Consequently, the sample fractions of our ice dh distribution**
**shown in Table 1 match the fractions of a fitted t distribution much better than the ones of a fitted**
**normal distribution. Reading example for |dh|>20m: only 0.03% of normally distributed samples**
**but 1.36 and 1.09% of the samples following a fitted t distribution and our dh distribution,**
**respectively, exceed the threshold of |dh|>20m.**
**We included a reference to Figure 2 in section 3.1 and hope that statistically interested readers will**
**be able to see the typical shape of a t distribution (of both *ice* and *land* samples) with their**
**relatively larger number of outliers in this figure themselves.**

**Table 1:** Example for the "heavier tails" of our sample distribution, as compared to a normal distribution. The numbers
correspond to the fraction of samples (area under the tails of the curve) for |dh| exceeding the threshold values 5, 10 and
20m (symmetric against zero) for a fitted normal distribution, fitted t distribution, and the original ice sample distribution.

| \|dh\|> | normal | t | samples |
|---|---|---|---|
| 5 | 0.4026 | 0.2345 | 0.2434 |
| 10 | 0.0773 | 0.0654 | 0.0760 |
| *20* | *0.0003* | *0.0136* | *0.0109* |

[Figure]

**Figure 7:** dh distribution of *ice* samples and corresponding t- and normal distributions fitted to the data.

*13. p6: did you experience any issues in the LIDAR data due to not fully adjusted flight*

*strips? (Remaining errors after strip-adjustment)*

**The LiDAR data was not used for glacier trends, in that sense the answer is no. Figure 4 in the manuscript shows that the LiDAR DEM elevations are closest to ICESat elevations, as compared to the Kartverket/SRTM DEM elevations, but an uncertainty remains nevertheless – which stems from uncertainties in ICESat elevations and, to some degree, likely also from remaining errors after strip adjustment. The sample number of only 184 LiDAR DEM autumn campaign samples over Hardangervidda is rather small and we could not detect a significant systematic spatial bias for any of the strips, or parts thereof, from splitting these samples into spatial subsets.**

*14. p6: what are possible reasons for the shifts in the Kartverkets DEMs?*

**We prefer to not discuss this further in the paper as the topic lies too much outside of the focus of this study. The main point is that the individual source units of ALL composite DEMs will in general not be perfectly aligned for various reasons depending on the DEM type and processing. In our case of the Kartverk DEM, we believe potential reasons for the shifts can be found in the general production processes of such national DEMs, at least older ones:**

- **Norway is a big/long and sparsely populated country and maps (and thus DEM units) have to be compiled from different air photo series, collected and processed over a range of years, with a range of equipment both for data acquisition and photogrammetric compilation.**
- **All these air photo blocks are adjusted individually and connection to pre-existing blocks cannot be perfect unless the air photos over the entire country are adjusted as one block.**
- **The air photo blocks and maps generated from them were compiled many years ago, without or with only little computer methods available and thus reduced consistency over large areas.**
- **The accuracy demands at the time of map production (and thus production of the contours that lie behind much of the DEM) were much lower than today, and it is actually well possible that they fulfilled these requirements at the time of production. In our study we compare a modern high-precision data set (ICESat) to old or very old reference elevations.**

*15. Section 4.1: how do you know the dh are t-distributed?*
**Visually, the dh distributions seem too narrow but have a larger number of outliers as compared to**
**a normal distribution (Figure 7). Statistically argued, the assumption of a t distribution seems**
**legitimate considering that a) the population standard deviation (of dh) is not known, and b) the**
**measured dh correspond to the sum of the measured glacier surface elevation change (the signal)**
**and multiple errors from both the DEM and ICESat as well as topography/clouds. Data with**
**additive errors/resulting from additive processes are more likely to have outliers (compared to**
**normally distributed data). In such cases regression based on a t-fit is suggested as a robust**
**method (Lange et al., 1989).**
**We tested the data both for normal and t-distribution. The assumption of a normal distribution of**
**our *ice/land* dh is rejected by an Anderson-Darling test (distribution fitted to the data). Both an**
**Anderson-Darling test for a t-distribution fitted to the data as well as a Kolmogoroff-Smirnoff test**
**in combination with a parametric bootstrap procedure to find a consistent estimate of the critical**
**value (the standard critical values from tables are not valid if the test distribution is estimated**
**from the data; Babu and Rao, 2004) are not able to reject the null hypothesis of a t-distribution.**
**While that is no proof, it gives us as much confidence as possible for the assumption of a t-**
**distribution of our data.**
*16. Saturation may occur along track when ICESat hits bare ice after rock (as it takes the gain _5*
*shots to reset after hitting the more reflective ice). Did you consider the spatial distribution of the*
*saturated waveforms? (compare Molijn RA, Lindenbergh RC, Gunter BC. ICESat laser full waveform*
*analysis for the classification of land cover types over the cryosphere. International journal of remote*
*sensing. 2011 Dec 10;32(23):8799-822)*
**The reviewer's comment relates to the automated gain loop built in into the data acquisition which**
**dynamically adjusts the gain based on received pulse intensities of the past laser shots (NSIDC,**
**2012). We think the reviewer has an important point here, given that a systematic bias at land/ice**
**transitions would mainly affect parts of the glacier where we expect larger dh (assuming glacier**
**melt/retreat). If this were the case it could affect and bias our trends.**
**The adjustment time of 5 shots the reviewer mentions corresponds to a ground track distance of ca.**
**860m. Many mountain glaciers (also in southern Norway) are smaller/narrower than this. While**
**we believe this might be more pronounced with larger ice bodies, our studies indicate that the**
**dynamic gain adjustment indeed seems very continuous in mountainous areas. There, the small**
**mountain glaciers and rough topography never really allow the sensor to settle for a certain gain.**
**In southern Norway we cannot see any pattern in the spatial distribution of the saturated samples,**
**also not where ICESat is passing over glacier margins and experiences a land/ice surface type**
**change. We don't see a consistent spatial pattern of the samples' gains either but find that as**
**many as 40% of all samples have the maximum gain of 250, independent of their saturation flag.**
**We also find that samples flagged as saturated have higher gains than non-saturated samples, both**
**for *ice* and *land* samples – this is the contrary of what one instinctively would expect. Interesting is**
**also that the gains of both *land* and *ice* samples increase with time, something that is beyond the**
**scope of this validation study but could potentially be relevant for other studies where**
**waveforms/surface types are the focus.**

**In this context, we would like to stress that the algorithm that flags samples as saturated is**
**designed for the flat ice sheet surfaces. Rough mountain surfaces – including the steeper and more**
**crevassed mountain glaciers – result in entirely different waveforms than laser returns from ice**
**sheets. From our experience, we don't find the saturation flag to be a good indication of saturation**
**– i.e. elevation bias – over mountainous topography. This finding, which is also in line with what**
**Kääb et al. (2012) found, is supported by the NSIDC GLAS user guide that doesn't generally**
**recommend saturation correction for the GLAH14 product (NSIDC, 2012), and which we refer to in**
**section 2.2.**
**We added a reference to Molijn et al. (2011) in section 5.3 to make readers aware of the possibility**
**of bias from saturated waveforms at land/ice transitions in other areas than southern Norway.**
**Given that we can't draw any useful conclusions or improvements from saturation analyses in our**
**study we hope the reviewer agrees with our decision not to discuss saturation (or sample gains) in**
**more detail in the manuscript resubmission.**
*17. You state: "However, these differences cancel out (Fig4)". Could you help the reader seeing that*
*in Figure 4.?*
**We rewrote the corresponding paragraph in section 4.3 to hopefully make the argumentation**
**clearer.**
*18. From the material just in this paper it is difficult to understand what you mean by p10r25-27:*
*"This stresses...weight". Could you explain this a bit more extensively?*
**The reviewer's comment concerns our statement that trends should be computed from individual**
**dh and not campaign medians. With this we tried to account for a question that we often hear**
**from the scientific community. Further explanations were added to create a better context in the**
**corresponding paragraph in section 4.4.**
*19. p12r6: you say "terrain characteristics" are essential, but, as argued by me before you only*
*consider these only in a very global way.*
**We are not sure what the reviewer would like us to change in that context. We found that "terrain**
**characteristics that govern glacier behaviour" need to be well represented – which is also how the**
**sentence in question is formulated in section 5.1. Thus, in the given context we refer to larger scale**
**terrain, not within-footprint terrain.  To be clearer about the scale of the terrain we refer to, we**
**also added the term "topography" in the corresponding sentence in section 5.1.**
*20. Section 5.3: do you believe that indeed the age of the DEM is crucial, or rather the way it was*
*constructed (photogrammetry, radar, LIDAR)?*
**This is very different for glaciers (surface elevation varies over time) and stable terrain. In general,**
**vertical bias is a result from mosaicking of different datasets which have (different) elevation bias.**
**The reasons for such bias are manifold and to a large degree depending on the quality of the data**
**acquisition and post-processing (see above response).**
**On glaciers, however, it is primarily the age of the reference DEM which is crucial in this context:**
**The glacier surface elevations of different years are expected to be different depending on**
**annually changing mass balance. Even under the assumption that the majority of the glaciers in**
**southern Norway follow the same cumulative mass balance curve (since they sit in the same**

**climate), reference elevations from different dates for different glaciers are not spatially uniform.**
**Combining these with ICESat's spatially varying sampling can potentially lead to severe bias. This is**
**why c_glac is more powerful – and more important – for glaciers than for the surrounding stable**
**terrain.**
**We added a paragraph in section 5.3 to better stress the important difference of this correction for**
*ice* **and** *land* **samples, respectively.**
*21. p15r13: saturation correction (and other flags). I would say this is an interesting topic for more*
*study, to check how the rapid transitions between land-cover on small mountain glaciers influence the*
*ICESat raw signal (and its corrections)*
**We agree with the reviewer that this could be an interesting topic for a study that focuses more on**
**single footprints and also assesses their waveforms. From the findings of Molijn et al. (2011) we**
**cannot exclude that there is a potential for a systematic bias from waveform saturation at ice/land**
**transitions, even though we could not detect any such bias in our study area. A paragraph**
**discussing this possibility has been added to section 5.3. For a more detailed argumentation on this**
**matter we refer to our reply to point 16.**
*22. p15: quality of the reference DEM vs ICESat: should it not be only the quality, but also its spatial*
*resolution compared to the ICESat footprints compared to the local relief variations?*
**We fully agree with the reviewer. We added a sentence to section 5.3 to make this clearer.**

## References

Andreassen, L. M., Elvehøy, H., Kjøllmoen, B., and Engeset, R. V.: Reanalysis of long-term series of
glaciological and geodetic mass balance for 10 Norwegian glaciers, Cryosphere, 10, 535-552,
10.5194/tc-10-535-2016, 2016.

Babu, G. J., and Rao, C. R.: Goodness-of-fit tests when parameters are estimated, Sankhya, 60 (1), 63-
74, 2004.

Gardner, A. S., Moholdt, G., Cogley, J. G., Wouters, B., Arendt, A. A., Wahr, J., Berthier, E., Hock, R.,
Pfeffer, W. T., Kaser, G., Ligtenberg, S. R. M., Bolch, T., Sharp, M. J., Hagen, J. O., van den Broeke,
M. R., and Paul, F.: A Reconciled Estimate of Glacier Contributions to Sea Level Rise: 2003 to 2009,
Science, 340, 852-857, doi: 10.1126/science.1234532, 2013.

Huss, M.: Density assumptions for converting geodetic glacier volume change to mass change,
Cryosphere, 7, 877-887, doi: 10.5194/tc-7-877-2013, 2013.

Ke, L., Ding, X., and Song, C.: Heterogeneous changes of glaciers over the western Kunlun Mountains
based on ICESat} and Landsat-8 derived glacier inventory, Remote Sens. Environ, 168, 13 - 23, doi:
10.1016/j.rse.2015.06.019, 2015.

Kropáček, J., Neckel, N., and Bauder, A.: Estimation of Mass Balance of the Grosser Aletschgletscher,
Swiss Alps, from ICESat Laser Altimetry Data and Digital Elevation Models, Remote Sens., 6, 5614,
doi: 10.3390/rs6065614, 2014.

Kääb, A., Berthier, E., Nuth, C., Gardelle, J., and Arnaud, Y.: Contrasting patterns of early twenty-first-
century glacier mass change in the Himalayas, Nature, 488, 495-498, doi: 10.1038/nature11324,
2012.

Kääb, A., Treichler, D., Nuth, C., and Berthier, E.: Brief Communication: Contending estimates of
2003-2008 glacier mass balance over the Pamir-Karakoram-Himalaya, Cryosphere, 9, 557-564, doi:
10.5194/tc-9-557-2015, 2015.

Lange, K. L., Little, R. J. A., and Taylor, J. M. G.: Robust Statistical Modeling Using the T-Distribution,
Journal of the American Statistical Association, 84, 881-896, doi: 10.2307/2290063, 1989.

Molijn, R. A., Lindenbergh, R. C., and Gunter, B. C.: ICESat laser full waveform analysis for the
classification of land cover types over the cryosphere, Int. J. Remote Sens., 32, 8799-8822, doi:
10.1080/01431161.2010.547532, 2011.

Neckel, N., Kropáček, J., Bolch, T., and Hochschild, V.: Glacier mass changes on the Tibetan Plateau
2003–2009 derived from ICESat laser altimetry measurements, Environ. Res. Lett., 9, 014009, doi:
10.1088/1748-9326/9/1/014009, 2014.

NSIDC: GLAS Altimetry HDF5 Product Usage Guide, NASA DAAC at the National Snow and Ice Data
Center, Boulder, Colorado USA, 2012.

NVE: Climate indicator products, Norwegian Water Resources and Energy Directorate, available at:
http://glacier.nve.no/viewer/CI/, accessed 31 May 2016, online glacier database, 2016.

Schutz, B. E., Zwally, H. J., Shuman, C. A., Hancock, D., and DiMarzio, J. P.: Overview of the ICESat
Mission, Geophys. Res. Lett., 32, doi: 10.1029/2005gl024009, 2005.

Shi, J. C., Menenti, M., and Lindenbergh, R.: Parameterization of Surface Roughness Based on
ICESat/GLAS Full Waveforms: A Case Study on the Tibetan Plateau, Journal of Hydrometeorology,
14, 1278-1292, doi: 10.1175/Jhm-D-12-0130.1, 2013.